# Selection-Inference: Exploiting Large Language Models for Interpretable Logical Reasoning

**Antonia Creswell, Murray Shannahan & Irina Higgins**
DeepMind,
London, UK
`{tonicreswell, mshanahan, irinah}@deepmind.com`

## Abstract

Large language models (LLMs) have been shown to be capable of impressive few-shot generalisation to new tasks. However, they still tend to perform poorly on multi-step logical reasoning problems. Here we carry out a comprehensive evaluation of LLMs on 46 tasks that probe different aspects of logical reasoning. We show that language models tend to perform fairly well at single step inference or entailment tasks, but struggle to chain together multiple reasoning steps to solve more complex problems. In light of this, we propose a Selection-Inference (SI) framework that exploits pre-trained LLMs as general processing modules, and alternates between selection and inference to generate a series of interpretable, casual reasoning steps leading to the final answer. Focusing on a sub-set of 10 reasoning tasks from ProofWriter and bAbI, we show that a 7B parameter, decoder-only LLM used within the SI framework in a 5-shot generalisation setting,with no fine-tuning, yields a performance improvement of over 100% compared to an equivalent Vanilla baseline. The same model in the same setting even outperforms a significantly larger 280B parameter baseline on the same suite of tasks. Moreover, answers produced by the SI framework are accompanied by a *causal* natural-language-based reasoning trace, which has important implications for the safety and trustworthiness of the system.

## 1 Introduction

Large language models (LLMs) are powerful few-shot learners (Bommasani et al., 2021; Brown et al., 2020; Lu et al., 2022). However, one area where they tend to perform poorly is logical reasoning (Rae et al., 2021). Yet the ability to perform multi-step, logically valid reasoning is fundamental for the discovery of new knowledge and explainability. It underpins many advancements that have been made in science, medicine, maths and philosophy. It is also one of the most valued strengths of classical, symbolic AI over contemporary deep learning methods (Marcus & Davis, 2019; Marcus, 2020; Bengio et al., 2021), prompting the recent increase in the use of neurosymbolic approaches to bridge this gap (Garnelo & Shanahan, 2019; Garcez & Lamb, 2020). Here we propose a Selection-Inference (SI) framework that takes inspiration from the neurosymbolic literature to improve the ability of LLMs to do logically valid reasoning.

There are many flavours of neurosymbolic models (Garcez & Lamb, 2020). Those from which we draw inspiration tend to have a modular structure, where each module is specialised for one type of operation (Mao et al., 2019; Andreas et al., 2016). For example, such modules may be neural networks or hand-crafted functions designed to attend to a single object, or to compare the location or size of two inputs (Andreas et al., 2016; Yi et al., 2018). Neurosymbolic models can produce an answer to a complex query by chaining these operations together, passing inputs from one module to another. This has the benefit of producing an interpretable trace of intermediate computations, in contrast to the "black-box" computations common to end-to-end deep learning approaches. Importantly, the modularity of neurosymbolic methods allows them to generalise to significantly harder problems that require long chains of reasoning (Hudson & Manning, 2019). However, the hand-crafted and

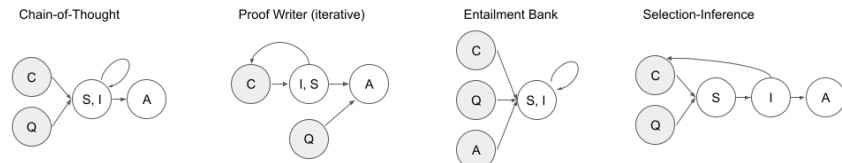

Figure 1: Schematic comparison between Selection-Inference and other representative approaches for reasoning in natural language: Chain-of-Thought (Wei et al., 2022), ProofWriter (Tafjord et al., 2021), and EntailmentBank (Dalvi et al., 2021). C - context, Q - question, A - answer, S - selection, I - inference. Grey circles - givens, white circles - model output. Loops indicate multi-step reasoning. The order of letters in a single circle indicates the order in which the corresponding steps are output by the model.

specialised nature of the modules often makes the resulting systems brittle and more difficult to extend to new domains (Yi et al., 2018). We hope to address this limitation by leveraging LLMs in our modules.

Building on other modular (Tafjord et al., 2021; Andreas et al., 2016) and step-wise approaches (Wei et al., 2022; Dalvi et al., 2021), we propose SI which decomposes logical reasoning into two modular stages: 1) *selection*, which involves choosing a subset of relevant information sufficient to make a single step of inference, and 2) *inference*, which only sees the limited information provided by the selection module, and uses it to infer a new intermediate piece of evidence on the way to the final answer (see Fig. 1). We implement both stages using pre-trained LLMs which, thanks to their powerful few-shot generalisation capabilities, serve as more general alternatives to the hand-crafted, specialised modules typically used in neurosymbolic approaches. In the SI framework, multiple steps of selection and inference are chained together to produce a sequence of reasoning steps. As well as underpinning better performance on reasoning problems, this yields an interpretable trace that justifies the final answer.

Furthermore, the reasoning trace produced by our system is *causal*, in the sense that each step follows from, and depends on, the previous step. Each inference step is made in isolation, based solely on the limited information provided by the Selection module, without direct access to the question or to previous steps of reasoning. This contrasts with the more common approach of obtaining *post-hoc rationalisation*, where the answer produced by the model has no direct dependence on the explanation, since the explanation is produced either in parallel to the answer or after the fact (Saha et al., 2020; Lampinen et al., 2022; Cobbe et al., 2021). A notable example that sits in the grey area between post-hoc rationalisation approaches and the more causal explanation approaches is Chain-Of-Thought (COT) (Wei et al., 2022) (see Fig. 1). In this approach LLMs are encouraged to produce a reasoning trace before the answer. However the dependence of the answer on the reasoning is not explicitly encouraged to be causal (as defined above). Indeed, the authors show that while the COT explanations help boost the final answer accuracy, the reasoning traces produced by the model are often wrong even when the final answer is correct (see the appendices of Wei et al. (2022) for examples) and the model is prone to making up facts (Figure A6). Developing a system that can demonstrate how it reaches its answers using a *causal* reasoning trace has important benefits in terms of safety, explainability, interpretability, debugging, and trust. In this paper we make the following contributions:

1. We provide a comprehensive evaluation of LLMs on a set of 46 tasks probing different aspects of logical reasoning, and show that LLMs are good at simpler single step logical inference in 5-shot generalisation settings, but struggle with harder problems (Sec. 3)

2. We introduce the Selection-Inference (SI) framework, a modular, iterative approach to solving reasoning problems (Sec. 4).

3. We demonstrate the utility of the SI framework by evaluating a 7B parameter, decoder-only, LLM[1] on 10 logical reasoning tasks, showing overall that it almost triples the performance of the same model used naively and almost doubles the performance of the same model used in the COT framework. Moreover, it often outperforms a 40x larger 280B, decoder-only, LLM baseline used both naively and in the COT framework.

4. We illustrate further benefits of the SI framework in terms of the causal and interpretable reasoning traces produced (Sec. 5). These traces can help humans understand how the model reached its final answer, which is useful for debugging and opens the system's decisions to human critique.

---

[1] Language model references throughout the paper are removed to preserve anonymity. These will be added back if the paper is accepted for publication. Note that we use decoder-only language models.

(a) Correct reasoning on bAbI deduction (left) and induction (right) tasks.

(b) SI can recover from an error (left) and justify an ambiguous answer with a reasoning trace (right).

Figure 2: Qualitative results from the Selection-Inference (SI) model on bAbI tasks.

## 2 RELATED WORK

Our Selection-Inference framework sits at the intersection of classical, symbolic AI and deep learning. A typical symbolic AI system might consist of a knowledge base, which is typically hand-curated by experts, and an inference engine that allows the system to perform logic-based reasoning over its knowledge base. One of the primary benefits of symbolic AI systems over deep learning models is their interpretability; we can look at the reasoning steps such a system has taken to see how the final conclusion was reached. However, unlike deep learning approaches, symbolic AI systems require knowledge to be hand-crafted and are in general hard to scale. Although some approaches have attempted to bridge the gap between deep learning and symbolic AI by converting problems into formal logic (Nye et al., 2021) and using existing solvers to help produce an answer, this process can be brittle and tends not to scale well. Alternatively, Mao et al. (2019); Yi et al. (2018); Gupta et al. (2019); Hudson & Manning (2019) propose models that bridge this gap by combining the best parts of deep learning – learning knowledge from data – and symbolic AI – producing an interpretable reasoning trace.

Recent work has attempted to adapt large pre-trained language models, LLMs, to the task of logical reasoning. At a high level these can be split into three groups: 1) approaches that try to fine-tune LLMs to produce the final answer directly, keeping reasoning implicit (Betz et al., 2021; Clark et al., 2021); 2) approaches that encourage LLMs to produce reasoning explicitly but all reasoning steps are produced in one generative step (Nye et al., 2022; Zelikman et al., 2022; Jhamtani & Clark, 2020; Dalvi et al., 2021; Wei et al., 2022; Cobbe et al., 2021), with notable examples including EntailmentBank and Chain of Thought; and 3) approaches that use LLMs to produce each reasoning step one at a time (Tafjord et al., 2021; Bostrom et al., 2022) (e.g. ProofWriter). The latter is where our Selection-Inference framework sits (see Fig. 1). In general it was found that the approaches that incorporate explicit reasoning work better than those that only try to predict the final answer. However, although explicit reasoning helps improve the accuracy of the models, encouraging the models to produce multiple steps of reasoning in a single generative pass is not enough to make the models use reasoning in a causal manner. The authors found that the generated reasoning traces often contain unrelated or incorrect steps while still resulting in the correct answer (see examples in the appendices of Zelikman et al. (2022) and Wei et al. (2022) and Sec. J). CoT is also prone to making up facts while reasoning (Figure A6). Encouraging LLMs to generate each reasoning step one at a time, for example ProofWriter (Tafjord et al., 2021; Bostrom et al., 2022), is currently the most promising direction for achieving causal reasoning, and it is the approach we take in our paper. While ProofWriter is very impressive, unlike our approach it only works for "Prove this statement to be True/False" style questions, since it relies on enumerating all possible implications and checking whether the question statement or its negation are present in the inferred facts, which is also computationally expensive.

## 3 HOW WELL DO LARGE LANGUAGE MODELS REASON?

Past work has shown that LLMs are poor at logical reasoning (Rae et al., 2021), however the evaluation was done on a relatively small set of tasks, and was not systematic. In particular, here we are interested in

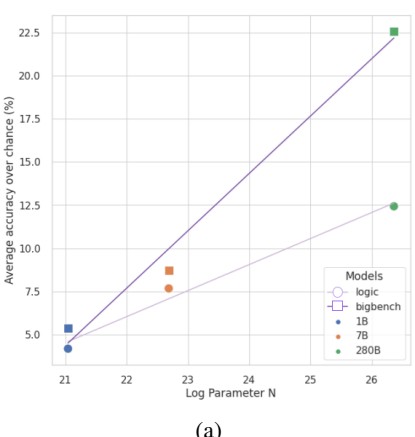
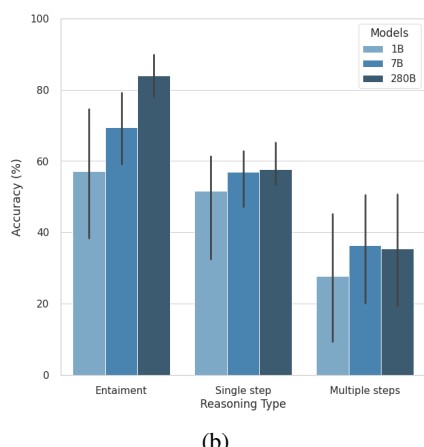

(a)                                                 (b)

Figure 3: Vanilla language models perform poorly on multi-step logical reasoning tasks. (a) Scaling laws for natural language tasks (bigbench, dark purple line, squares, 56 tasks) and tasks involving logical reasoning (logic, light purple line, circles, 38 tasks). All accuracy results are calculated relative to the random baseline (0% accuracy means chance level). Only multi-choice tasks are used. (b) Language models perform well for simple entailment tasks (AAC tasks, Entailed Polarity), their performance starts to get worse on single step inference problems (bAbI task 1, ProofWriter tasks 0-1), and they struggle with more complex multi-step reasoning problems (2WikiMultiHop tasks, bAbI tasks 2-3, ProofWriter tasks 2-5, StrategyQA). A detailed per-task breakdown is shown in Figure A5.

1) how LLMs perform on simple entailment tasks compared to multi-step reasoning problems and 2) how scaling laws apply to logical reasoning. To this end, we evaluated decoder-only LLMs of various sizes in a 5-shot[2] setting, following the same protocol use for the BigBench evaluation in Rae et al. (2021), on a larger set of 46 tasks. These tasks touch on different aspects of logical reasoning and vary in terms of the number of reasoning steps required, presence or absence of negation, whether the relevant context information was provided, and whether the model is required to evaluate the accuracy of multiple choices or generate the answer among others. The additional tasks were collected from six sources: bAbI (Weston et al., 2016), BigBench (Ghazal et al., 2017), AAC (Betz et al., 2021), Jeopardy (Tunguz, 2019), ProofWriter (Tafjord et al., 2021) and 2WikiMultiHop (Ho et al., 2020) (see Fig. A5a for raw results).

Our analysis found that LLMs are good at some aspects of logical reasoning (Fig. 3b). In particular, they appear to be good at simple entailment and implication tasks (e.g. see AAC tasks and Entailed Polarity in Fig. A5a). This appears to hold when negation is present (AAC Split Extended tasks), and both in generative (AAC Split) and multiple-choice scoring settings (AAC Split Extended tasks, Entailed Polarity). However, the performance of vanilla language models tends to decrease when they are presented with irrelevant facts alongside the ones relevant for reasoning (e.g. see 2WikiMultiHop With Context tasks, bAbI tasks 2-3 or ProofWriter tasks), when they have to infer the relevant facts from memory (e.g. 2WikiMultiHop or StrategyQA tasks), and as the questions start to require more steps of reasoning (e.g. see the performance drop between bAbI tasks 1-3 or ProofWriter Tasks).

Our results confirmed that LLMs of larger sizes do perform better than the smaller models. However, we found that even the largest 280B model performed only 13.6% above chance level on average across the 38 available multi-choice logic tasks (see Figs. A5a-A5b and Sec. H for more details). Furthermore, we found that logical reasoning tasks were qualitatively different from other natural language tasks. The scaling law for logic-based tasks was significantly worse than for other language tasks measured here as the average performance on the subset of BigBench tasks described in Sec. H. This subset included no logic tasks (see Fig. 3a).

## 4   THE SELECTION-INFERENCE (SI) FRAMEWORK

We are interested in solving logical reasoning problems expressed in natural language. In this work we assume that each question is accompanied by context information (see Fig. 2), which contains

---

[2]We chose 5-shot setting because Min et al. (2022) have demonstrated that additional shots beyond 5 result in limited increase in multi-choice accuracy

all the information necessary to solve the problem, as well as potentially irrelevant distractors. In the future this assumption can be relaxed, for example by extracting the necessary information through search (Lazaridou et al., 2022; Menick et al., 2022). We also assume that all questions are well posed and definitively answerable given the context.

Logical reasoning problems require using existing information to infer new relevant knowledge necessary to answer the question. This can be done through deduction, induction or abduction, although the datasets we use here contain mostly deductive and a small number of inductive problems[3]. Some problems require multiple steps of inference, where later steps use the knowledge inferred in the earlier steps. Hence, we use an iterative framework where at each step the SI uses information in the existing context, $\mathcal{C}_t$, to infer a new fact, $f_t$, which is appended back to the context to create new context, $\mathcal{C}_{t+1} = \mathcal{C}_t \cup f_t$. This process can then iterate until the solution to the question is found. In the current implementation of the SI framework, we repeat the process for a fixed number of steps and take the final inference to be the answer. Addressing the issue of halting is left for future work.

Inspired by neurosymbolic methods, we additionally split each step of reasoning into further two components: 1) *Selection*, which selects a subset of the information present in the context, $s_t$, given the context and the question, $\mathcal{C}^t \cup q$. This selection, $s^t$ is fed to the next step, 2) *inference*, which produces the new fact, $f_t$, based on the information passed to it by the selection step (Fig. 1). Examples of selection and inference are shown in Fig. 2. This splitting of each step of reasoning into selection and inference is the main contribution of our paper, and is important for several reasons. First, and most importantly, it makes the resulting reasoning *causal*, since both steps have limited capabilities by design, and are interdependent. The selection step is constrained to only use the information available in the context (similar to Gupta et al. (2022) in the tabular setting), and the inference step only sees the subset of facts provided by the selection without access to the question. Hence, the model is unlikely to make up information to answer the question, and it cannot ignore reasoning when producing the final answer. The other benefit of this approach is that each step of reasoning is broken down into even smaller sub-tasks, which are easier for LLMs to adapt to, and which helps make the reasoning more generalisable to harder problems.

In this paper we use pre-trained, frozen language models in a 5-shot generalisation setting using prompt engineering to implement the Selection and Inference modules. We settled on prompt engineering to evaluate the base utility of the SI framework, however it can also be used in the fine-tuning setting which we explore briefly in Sec. 6. We next describe the Selection and Inference modules in more detail.

## 4.1 SELECTION MODULE

We use prompt engineering to encourage the model to output the correct selection, $s^t$. The n-shot prompt is a string of the following form:

```
# n-shot prompt
# First example.
<context 1> \n <question 1>
# Example selection
<fact>. We know that <fact>[ and <fact>]*. Therefore,
...
# Problem to solve.
<context> \n <question>
```

where `<fact>`s are copied directly from the `context`, and `[ and <fact>]*` means that the module is allowed to select more than one fact for each step of inference, where the total number of facts is a hyper-parameter.

The simplest option to implement the selection is to feed this prompt directly to a pre-trained LLM and take the output generated by the language model. However, this unconstrained approach may allow the model to make up facts, thus removing the causal aspect of the reasoning trace. Indeed during experimentation this is what we often found. So instead we use the pre-trained LLM to score each of the facts in the context, and select the one with the highest log-likelihood. We can then repeat this process by appending each new fact to the end of the previous prompt until the full selection is constructed. Note that for now we `halt` after a fixed number of steps. See Algorithm 2 for more details.

---

[3]See Fig. 2 for an example of deduction and induction problems used in this paper.

---

**Algorithm 1** Selection-Inference

---

**Require:** An n-shot selection prompt, $p_{select}$.
**Require:** An n-shot inference prompt, $p_{infer}$.
**Require:** Initial Context, $\mathcal{C}^0$, made up of statements e.g. facts and rules.
**Require:** The question, $q$.
**Require:** Language model, LLM.
**Require:** The number of reasoning steps, $H$.
  $t = 0$                                                     ▷ Start at step 0.
  **while** $t < H$ **do**
    $s^t \leftarrow$ Selection_Module($p_{select}, C^t, q$, LLM)            ▷ Do selection.
    $i^t \leftarrow$ Inference_Module($p_{infer}, s^t$)                   ▷ Do inference.
    $C^{t+1} \leftarrow C^t \cup i^t$           ▷ Add the newly inferred fact to the context.
    $t \leftarrow t + 1$                    ▷ Move onto the next step of reasoning
  **end while**
  **return** $s^t$

---

## 4.2 INFERENCE MODULE

The n-shot prompt for the Inference module has the following form (shown below):

```
#n-shot inference prompt
# First example.
<fact>. We know that <fact>[ and <fact>]*. Therefore, <new fact>.
...
# Problem to solve.
<output of the Selection module>. Therefore,
```

The n-shot prompt and the output of the Selection module, are fed to the pre-trained LLM serving as the Inference module. The first generated sentence is taken to be the newly inferred fact. This fact is added to the context, which concludes one reasoning step of the SI framework. For now, we halt after a fixed number of steps. See Algorithm 1 for more details.

## 5 EXPERIMENTS AND RESULTS

We evaluate our SI framework on a subset of 10 /46 logical reasoning tasks introduced in Sec. 3. These tasks were chosen based on whether they include context information necessary to answer the question, whether the questions have a definitive answer, and to ensure that they cover different kinds of reasoning abilities. The tasks include bAbI (Weston et al., 2016) Tasks 1-3, which require the model to use 1-3 supporting time-ordered facts respectively to answer a question, and Tasks 15-16, which test deductive and inductive reasoning respectively. We also evaluate our model on a subset of the ProofWriter (Tafjord et al., 2021) OWA datasets that matches closed world assumptions, for depths 0, 1, 2, 3 and 5 (there is no depth 4 task, also see Sec. D.4 for an explanation for we did not use CWA dataset directly). These are language-based logical reasoning problems, where the depth is the number of reasoning steps required to answer the question.

To baseline the performance of the SI framework, we consider a 7B (same size as the LLM used in the SI framework) and a 40x larger 280B parameter LLM evaluated in a 5-shot setting. There are two types of evaluation for these vanilla baselines that we consider: *multi-choice* and *generative* evaluation. In *generative* evaluation, we measure the exact string match (first sentence in lower case and ignoring any non-alphabetic characters) between the output generated by the LLM and the ground truth answer. This is appropriate, since most of the tasks that we consider require either a single word answer, or the dataset is such that the answers are highly structured. In *multi-choice* evaluation the LLM is used to score each of the answer choices, as in Li et al. (2022). In general LLMs perform significantly better in a *multi-choice* vs *generative* evaluation setting, since the chance level in the multi-choice setting is significantly higher.

We also consider a chain-of-thought (COT) (Wei et al., 2022) inspired baseline, where the k-shot prompts to the 7B and 280B models include reasoning traces for the same examples that we use to prompt the SI framework (although with selection and inference combined, see Appendix B for example prompts). This tests whether providing the reasoning examples alone is sufficient to improve performance, or whether the further breakdown into Selection and Inference sub-steps improves

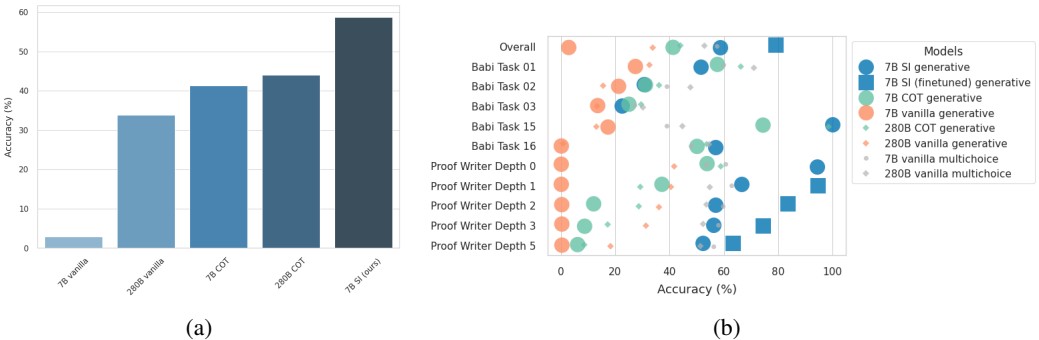

Figure 4: Quantitative results for the Selection-Inference (SI) framework. (a) Average accuracy over 11 datasets comparing like-for-like generative performance of the 7B and 280B parameter language models used in a 5-shot generalisation setting to predict the answer directly (vanilla), in the Chain-Of-Thought framework, COT, and in the SI framework. (b) Per task breakdown of the performance of LLMs used naively, and within the COT and SI frameworks.

accuracy. Note that among all of the approaches outlined only the SI framework is explicitly set up to generate *causal* reasoning traces.

Fig. 4a demonstrates that overall when evaluated generatively, the 7B parameter LLM within the SI framework performs better (58.75%) than the same model evaluated naively (2.94%) or in the COT framework (41.32%) (all $p < 0.01$, see Appendix I for details of the significance test calculations). Not only that, the 7B parameter LLM within the SI framework also outperforms on average the 40x larger 280B parameter LLM in both vanilla (33.84%) and COT framework (44.03%) (all $p < 0.01$). When evaluated in the easier multi-choice setting, we find that surprisingly[4] the vanilla 7B parameter LLM outperforms the 280B parameter LLM (59.21% vs 52.94%), while still performing significantly worse than the 7B SI model ($p = 0.012$). Note that the latter is evaluated in the harder generative setting. Per task breakdown shown in Fig. 4b demonstrates that the SI framework solves the bAbI 15 deduction task, the only model to achieve 100% accuracy (significant difference from the other models, $p < 0.01$). Furthermore, it does so having seen only five examples in the prompt. The 7B SI model also significantly outperforms all other models on ProofWriter Depth 0 ($p < 0.01$), ProofWriter Depth 1 ($p = 0.034$).

As well as improving upon most baselines quantitatively, the SI framework also has additional qualitative benefits: 1) it produces a causal, human interpretable reasoning trace that shows how the model reached its answer and 2) it is able to recover from errors. We will now discuss each of these in turn.

Since the Selection module is only allowed to pick facts from the context and is separate from the Inference module, and since the latter does not have access to the question, the model has to use what is selected and cannot bypass the selection to compute its answer, thus creating a causal reasoning trace. Human evaluation of the quality of the produced reasoning traces on bAbI tasks found that on average SI produced more valid selections (60% vs 47%), and more correct inferences (56% vs 38%) than CoT (see Tbl. A1 for more details). Since the reasoning trace is in natural language and is causal, it can be audited, inspected and corrected by humans, which has significant implications for safety and interpretability (see demo video in Supplementary Materials).

Example reasoning traces produced by the SI model are shown in Fig. 2. In the bAbI 16 example shown on the right the model is solving an inference problem, which requires the model to infer the colour of an animal given facts about the colours of other animals. In this example, the model is asked `"What colour is greg"`, and told that `"greg is a lion"`. This means first the model needs to use induction to infer a rule about the colour of lions. On the first step, we see that the model induces a rule, `"lions are white"`, based on the fact that `"brian is a lion"` and `"brian is white"`; we can see exactly what data underlies the model's decision to form a new rule. On the second step, we see that the model applies this newly inferred rule to the fact that `"greg is a lion"` to reach the final conclusion that `"greg is white"` using deduction. Note that the ability of the SI framework

---

[4]This *could* suggest that the 280B LLM has stronger priors, than the 7B LLM, which it favours over logical reasoning. For example, favouring `sheep are afraid of wolves` despite a context to the contrary (Min et al., 2022). However this requires further investigation.

**Context:**
a toaster is made of a heating element for toasting
a heating element converts electrical energy/electricity into heat energy for heating
Question: What energy change takes place when a piece of bread is toasted in a toaster? chemical energy to light energy OR electrical energy to heat energy OR heat energy to chemical energy OR light energy to electrical energy.

**Selection:** a toaster is made of a heating element for toasting. We know that a heating element converts electrical energy/electricity into heat energy for heating
**Inference:** the toaster converts electrical energy into heat energy for toasting
**Correct Answer:** a toaster converts electrical energy to heat energy for toasting

**Context:**
adaptations are used for survival
an animal can survive in an environment with little food by storing fat
a camel's humps are used for storing fat
Question: Camels have humps on their backs that store fat, allowing them to survive for many days without food. This makes camels well suited to desert life. This characteristic is an example of adaptation OR instinct OR migration OR hibernation

**Selection:** adaptations are used for survival. We know that an animal can survive in an environment with little food by storing fat
**Inference:** an example of an adaptation is storing fat for times when food is scarce

**Selection:** an example of an adaptation is storing fat for times when food is scarce. We know that a camel's humps are used for storing fat
**Inference:** a camels humps are a kind of adaptation for storing fat when food is scarce
**Correct Answer:** an example of an adaptation is camel humps storing fat

Figure 5: Handpicked example solutions obtained from the Selection-Inference framework on the EntailmentBank dataset. Left: one-step reasoning; right: two-step reasoning. See Sec. M for more examples which have not been handpicked.

to produce inductions relies on its ability to deal with uncertainty and understand what is "reasonable" - something that LLMs are naturally capable of, while also being a known struggle point for symbolic AI.

Since the reasoning traces are output in natural language, they are easy for humans to interpret and potentially intervene. Consider a scenario where there may be both a white lion and a green lion mentioned in the context, in which case we could see which information the model used to make its final decision and decide whether we want to trust it (example in Fig. 2b). We could also imagine examples where the model puts together two unrelated facts to come up with an incorrect inference, and this could also be easily be spotted by a human and rectified by replacing the wrongly inferred fact with a correct one, and re-running the consequent reasoning steps.

Aside from inspecting reasoning traces and using them to debug when something goes wrong, the additive nature of our model - it accumulates new knowledge with each reasoning step, means that it also has the ability to recover from errors. Fig. 2b demonstrates such an example. In the first step the model inferred that `"swans are often gray"`, using the facts that `"julius is a swan"` and `"julius is gray"`. While this is correct, this new information is not useful for answering the question, which asks about lions. However, it is still possible for the model to make the more useful inference that `"lions are often white"` in a later step and recover from its original misstep.

# 6 Fine-tuning Language Models for Selection and Inference

In Sec. 5 we have demonstrated significant improvements in logical reasoning accuracy when using prompt-engineering to specialise LLMs for Selection and Inference in the SI framework. Prompt-engineering has the additional benefit of not requiring large amounts of step-by-step reasoning data, which may be hard to obtain. In this section we investigate whether the accuracy of the SI framework can be further improved by fine-tuning the LLMs for Selection and Inference. We present extensive results on the ProofWriter dataset. Finally, to show that our approach also works on more complex natural language tasks we show qualitative results on science problems from EntailmentBank Dalvi et al. (2021). Both ProofWriter and EntailmentBank datasets provide ground truth reasoning traces.

The Selection LLM is fine-tuned to select a subset of sentences (including facts and rules) from the context by generating a string of the form *"sent 2. We know that sent 4 [and sent 7]*."* given the context and the question. We ask the Selection LLM to generate sentence labels (e.g. *"sent 2"* or *"sent 4"*) instead of the sentences themselves, because this prevents the Selection LLM from cheating by making up facts to answer the question quicker. This preserves the dependency of the selection and therefore subsequent reasoning steps on the context. The Inference LLM is fine-tuned to compute an entailment given the selection. Both models are fine-tuned on single steps of reasoning only. Example training data are shown in Fig. A2.

On the ProofWriter dataset, the Inference LLM converged very quickly to >99% test accuracy after only 300 fine-tuning steps with a batch size of 16, which is to be expected as we found that pre-trained LLMs are good at single step entailment out of the box as shown in Fig. 3b. Examples of inferences made by the model are shown in Fig. A3. The Selection LLM was trained for $4 \times 10^4$ steps (with batch size 16 for 50 hours on a TPU) with the exact string match accuracy reported in Fig. 6a. Although we notice that the model is much better at predicting selections for problems that require fewer steps of inference than those that require more, ultimately the model still achieves high ($>80\%$) accuracy

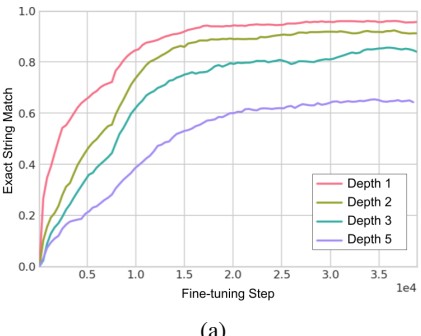 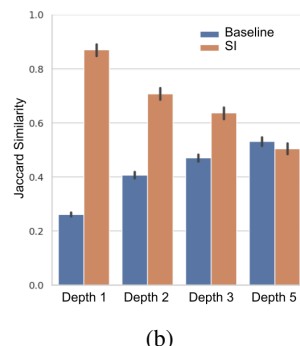

(a) (b)

Figure 6: Fine-tuning the SI framework on the ProofWriter dataset. (a) Average test fine-tuning accuracy for the Selection module trained on single-step selection across all ProofWriter datasets (depth 1, 2, 3 and 5). (b) Intersection over union between reasoning traces produced by a model and the ground truth reasoning steps. Baseline, 7B parameter LLM fine-tuned to predict all reasoning steps in one go; SI, using same 7B LLM fine-tuned for single step Selection and Inference.

across most of the reasoning depths. Predicting early selections for deeper reasoning problems is hard, because it requires planning. It is an important problem to address in future work.

Fig. 4b shows that fine-tuning LLMs on single steps of reasoning within the SI framework leads to significant improvements in final reasoning accuracy (78.95%) over the prompt-engineered version of the SI framework (57.93%) and other prompt-engineered baselines (vanilla/COT generative 7B: 0.34/15.73%, 280B: 31.58/21.12%). We also found that the fine-tuned 7B LLM used within the SI framework produces significantly more accurate reasoning traces compared to the same LLM fine-tuned to predict all reasoning steps in one go (Fig. 6b). We quantified this using the Jaccard Similarity, `Jaccard Similarity` $= (M \cap GT)/(M \cup GT)$, between the proof steps predicted by each model, $M$, and the ground-truth reasoning steps, $GT$, as shown in, calculated using exact string match over alphanumeric characters. Qualitatively we observed that while the baseline model is good at predicting most of the reasoning steps, they often appear in the wrong order, there are additional reasoning steps that are not on the minimal reasoning path, and some steps get repeated a number of times. Fig. 5 and Sec. M), demonstrate that our approach is able to generalise beyond procedurally generated datasets and can reason about the natural science questions too.

## 7 CONCLUSION

We have presented the Selection-Inference framework for improving the ability of pre-trained language models to solve logical reasoning problems expressed in natural language. Our approach borrows from the best practices of neurosymbolic approaches to break down logical reasoning into a modular recursive pipeline that not only significantly improves the reasoning accuracy, but also produces a *causal* interpretable reasoning trace. We have demonstrated that prompt-engineered LLMs used in the SI framework significantly outperform both the vanilla and COT baselines evaluated in equivalent settings, and even 40x larger baselines. The performance of the SI framework can be further improved through fine-tuning if step-by-step reasoning data is available.

A model capable of casual, interpretable and logically valid multi-step reasoning has potential applications in law, medicine, science, maths, economics, and other areas where trustworthy and verifiable logical inference is important. At the same time we recognise that special care will need to be taken to evaluate such a model before deployment in any of these settings. Further work is also needed, for example, to improve the Selection module (e.g. by allowing the model search over and evaluate different reasoning traces); to address the halting issue (both in terms of when to stop the selection and when to stop the overall reasoning); to incorporate verifiers that would help avoid false inferences being added to the context; to enable the system to source its own relevant context rather than relying on it being provided in the dataset; and to extend the ability of the system to deal with ambiguous or unanswerable questions.

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

# Appendix

## Table of Contents

## A  EXAMPLE PROMPTS FOR VANILLA BASELINES

In this section we show example prompts used to obtain the Vanilla language model baselines. For each sample, a 5-shot prompt is composed by sampling five (question, answers) pairs from the batch and composing them according to a simple template. See examples below.

### A.1  PROOFWRITER

```
"""
Here are some statements that describe a situation:
Bob is cold.
Charlie is quiet.
Gary is cold.
Harry is quiet.
Big things are cold.
All blue things are not cold.
If something is quiet and blue then it is not cold.
All quiet things are cold.
If something is big and rough then it is round.
If something is cold and not rough then it is blue.
If something is quiet and not furry then it is not blue.
Round things are big.
Based on the above, the statement "Charlie is cold" is true.

...

Here are some statements that describe a situation:
Erin is not cold.
Erin is kind.
Erin is red.
Erin is smart.
Erin is not white.
Erin is young.
Gary is cold.
Gary is not furry.
Gary is kind.
Gary is red.
Gary is not smart.
Gary is young.
All cold, smart things are not furry.
Young, cold things are not furry.
If something is white and smart then it is furry.
If Gary is white then Gary is not furry.
If Erin is young then Erin is furry.
If Gary is not young then Gary is smart.
If Erin is cold then Erin is young.
Red things are kind.
Based on the above, the statement "Erin is not furry" is
"""
```

### A.2  BABI 1

```
"""
Context: daniel went to the bedroom
daniel journeyed to the office
daniel travelled to the bathroom
mary went to the office
john journeyed to the bedroom
```

```
daniel went back to the kitchen
john went to the garden
daniel travelled to the office
Question: where is john?
Choice: garden
Choice: bathroom
Choice: office
Choice: kitchen
Choice: bedroom
Choice: hallway
Answer: garden

...

Context: sandra went to the kitchen
sandra went to the office
sandra travelled to the hallway
sandra went back to the kitchen
mary travelled to the hallway
sandra went to the bedroom
john went to the garden
sandra travelled to the office
Question: where is sandra?
Choice: garden
Choice: bedroom
Choice: kitchen
Choice: bathroom
Choice: hallway
Choice: office
Answer:
"""
```

### A.3 2WIKIMULTIHOP

New lines are added between facts to fit on the page.

```
"""
    Q: When did Michael Baden-Powell's father die?
Here are some relationships to help answer this question.
Michael Baden-Powell::father::Peter Baden-Powell, 2nd Baron Baden-Powell,
Peter
    Baden-Powell, 2nd Baron Baden-Powell::date of death::9 December 1962
A: 9 December 1962

...

Q: Where does Ekaterina Rybolovleva's father work at?
Here are some relationships to help answer this question.
Ekaterina Dmitrievna Rybolovleva::father::Dmitry Rybolovlev,
Dmitry Rybolovlev::employer::Uralkali
A:
"""
```

## B EXAMPLE PROMPTS FOR COT BASELINES

In this section we show example prompts used to obtain the Chain-of-Thought baselines.

### B.1 PROOFWRITER 3

```
"""
Given a set of rules
    and facts, you have to reason whether a statement is true or false.
Here are some facts and rules:
If someone is red then they are nice.
If someone is kind and red then they are white.
If someone is nice then they are kind.
Fiona is red.
Does it imply that the statement "Fiona is not white" is True?
Reasoning: If someone is red then
    they are nice. We know that Fiona is red. Therefore, Fiona is nice.
If someone is nice then
    they are kind. We know that Fiona is nice. Therefore, Fiona is kind.
If someone is kind and red then they are white. We
    know that Fiona is kind and Fiona is red. Therefore, Fiona is white.

...

Here are some facts and rules:
If someone chases the cow then they eat the cow.
If someone is big then they chase the cow.
If someone needs the bald eagle then the bald eagle is big.
If the
    bear is nice and the bear needs the cow then the bear eats the lion.
If someone needs the lion and they eat the bald eagle then they are blue.
If someone
    eats the bear and they do not chase the cow then the cow is young.
the bald eagle eats the lion.
the bear is round.
the lion eats the bald eagle.
the bald eagle needs the cow.
the bear is young.
the cow is not nice.
the cow does not chase the bald eagle.
the bear does not eat the bald eagle.
the bear needs the bald eagle.
the bald eagle chases the bear.
Does it imply
    that the statement "The bald eagle does not eat the cow" is True?
Reasoning:
    If someone needs the bald eagle then the bald eagle is big. We know
   that the bear needs the bald eagle. Therefore, the bald eagle is big.
If someone is big then they chase the cow. We know
   that the bald eagle is big. Therefore, the bald eagle chases the cow.
If someone chases the cow then they eat the cow. We know that the
    bald eagle chases the cow. Therefore, the bald eagle eats the cow.
"""
```

## B.2   BABI 2

```
"""
Below are some
    stories about people moving objects between rooms. After each story
    you have to answer a question about where a particular object is.
Story:
at t=0 mary grabbed the football there
at t=1 daniel got the apple there
at t=2 mary went to the kitchen
at t=3 daniel journeyed to the office
at t=4 daniel went to the bedroom
at t=5 mary moved to the garden
Question: where is the apple?
Reason: at t=1 daniel got the apple there. We know that at t
   =4 daniel went to the bedroom. Therefore, the apple is in the bedroom
```

```
...

Story:
at t=0 sandra went to the office
at t=1 john took the milk there
at t=2 sandra got the milk there
at t=3 john dropped the milk
Question: where is the milk?
Reason: at t=2 sandra got the milk there. We know that at
    t=0 sandra went to the office. Therefore, the milk is in the office
"""
```

## C  EXAMPLE PROMPTS FOR SELECTION-INFERENCE

In this section we show example prompts used for both the Selection and Inference models in the Selection-Inference framework.

### C.1  BABI 2

The **selection** prompt:

```
"""
Here are a collection
    of stories about people carrying objects from one room to another
    . You will be asked where any object is. To answer this question
     you need to figure out who last had the object and which room they
     have the object in by the end of the story. Here are some examples:

Story:
at t=0 mary grabbed the football there
at t=1 daniel got the apple there
at t=2 mary went to the kitchen
at t=3 daniel journeyed to the office
at t=4 daniel went to the bedroom
at t=5 mary moved to the garden
Question: where is the apple?
Reason: at t=1 daniel
    got the apple there. We know that at t=4 daniel went to the bedroom

...

at t=0 john moved to the bathroom
at t=1 john travelled to the office
at t=2 john picked up the football there
at t=3 john journeyed to the bathroom
Question: where is the football?
Reason:

"""
```

The **inference** prompt:

```
"""
at t=1 daniel got the apple there. We know that at t=4
    daniel went to the bedroom. Therefore, the apple is in the bedroom.

...

at t=2 john picked up the
    football there. We know at t=0 john moved to the bathroom. Therefore,

"""
```

## C.2 PROOFWRITER

Below is an example **selection** prompt. Note that this is for a depth-2 problem and so we show examples of the first reasoning step where the conclusion would not directly prove or disprove the statement and the last reasoning step, where the conclusion would directly prove or disprove the statement.

```
"""
Given a set of rules
    and facts, you have to reason whether a statement is true or false.

Here are some facts and rules:
Nice people are quiet.
If Dave is smart then Dave is nice.
All white people are smart.
Dave is smart.
Harry is cold.
Does it imply that the statement "Dave is not quiet" is true?
Reasoning: If Dave
    is smart then Dave is nice. We know that Dave is smart. Therefore,

Here are some facts and rules:
Blue things are green.
All blue things are white.
If Anne is not big then Anne is blue.
Big things are white.
All kind things are round.
If something is white and big then it is not kind.
If something is big and not rough then it is green.
If something is white and blue then it is not green.
Erin is not white.
Anne is big.
Bob is rough.
Anne is white
Does it imply that the statement "Anne is kind" is True?
Reasoning: If something is white and big then it
    is not kind. We know that Anne is white and Anne is big. Therefore,

...

Here are some facts and rules:
If something
    likes the squirrel and it is not young then it chases the lion.
If something likes the squirrel then it is rough.
If something chases
    the rabbit and the rabbit is not young then it chases the lion.
If something eats the lion then it is young.
If something likes the rabbit then it chases the rabbit.
All rough things are nice.
the rabbit is young.
the squirrel likes the rabbit.
the lion likes the squirrel.
Does it imply that the statement "The lion is not nice" is True?
Reasoning:
"""
```

Example **inference** prompt:

```
"""
Nice people
    are quiet. We know that Dave is nice. Therefore, Dave is quiet.

...
```

```
If the cow chases the bald eagle then the cow eats
    the bald eagle. We know that the cow chases the bald eagle. Therefore
"""
```

# D  SELECTION-INFERENCE EVALUATION DETAILS

## D.1  SELECTION MODULE

The algorithm for the Scoring Selection module is shown in Algorithm 2.

---
**Algorithm 2** Scoring `Selection_Module`

---
**Require:** An n-shot prompt, $p$.
**Require:** Initial Context, $\mathcal{C}^0$, made up of statements e.g. facts and rules.
**Require:** The question, $q$.
**Require:** Language model, LLM.
**Require:** The number of statements to select from the context, $K$.
  $s^t \leftarrow$ empty string
  **while** $t < K$ **do**

      $s_{\text{temp}} \leftarrow \text{argmax}_{\text{rule\_or\_fact} \in \mathcal{C}} \sum_{\text{token} \in \text{rule\_or\_fact}} \text{LLM}(\text{token}|p, \mathcal{C}^t, q, s^t)$    ▷ Choose the rule or fact with the maximum log-likelihood under the LM model.

      $s^t \leftarrow \text{join}(s^t, s_{\text{temp}})$          ▷ Join the selected fact or rule to the selection string.
  **end while return** $s^t$

---

## D.2  INFERENCE MODULE

To extract the new fact to be added to the context we filter out the first sentence of the text generated by the LLM using the following regular expression: `r'[^.?!; n]+'`.

## D.3  BABI

For all bAbI tasks, the answer is a single word. For example, in bAbI 1-3 the answer is one of `["hallway", "bathroom", "bedroom", "garden", "kitchen", "office"]`; for bAbI 15 the answer is one of `["sheep", "cat", "mouse", "wolf"]` and for bAbI 16 the answer is one of `["yellow", "gray", "green", "white"]`. However, our model outputs a complete sentence, for example `"emily is afraid of mice"`. Therefore, we take the answer to be the final word output by the inference model on its last step.

To obtain the results for bAbI tasks 1-3, 15 and 16 shown in Fig. 4b we prompted the language model to solve the problem in a single step of reasoning. An example of such a prompt is shown in Sec. C.1. We run the SI model for only a single step of reasoning too. Additional steps may increase the chance of the model reaching the correct answer, however, we do not yet have a mechanism for halting reasoning when the answer is reached.

BAbI 16 is an inductive reasoning task that *could* be solved in two steps (rather than one). The first step requires a rule to be inferred and the second step requires the inferred rule to be applied to another fact. For this reason, we also apply SI for two steps to solve the bAbI 16 problems, first inferring a rule from a number of facts and then applying the rule to the correct fact. An example of this is shown in Fig. 2. Using this two step approach, we can see exactly which facts contributed to the formation of a new rule.

## D.4  PROOFWRITER

We use a subset of the ProofWriter Open World Assumption, OWA, dataset. In the Close World Assumption dataset, CWA, everything that can be proven is True otherwise it is False. This means things are either True or False. This also means that reasoning traces are only provided when a statement

is True, but not when a statement is False. To "show" something is False one has to enumerate all possible facts (possibly up to a certain depth) and then if a statement is not shown to be True it is assumed to be False. It is therefore not simple to generate meaningful reasoning traces for these types of problems.

On the other hand, in the OWA data if it is not possible to prove something is True or False, then it is Unknown. This means that for True and False examples, where one may want to show `p(x)`, reasoning traces are available that terminate in `p(x)` (for True) or `not p(x)` (for False). If one cannot show `p(x)` or `not p(x)` then the answer is Unknown, and again there is not a clear reasoning trace for this; it is necessary to enumerate all possible facts (possibly up to a certain depth) and then if one has not shown `p(x)` or `not p(x)` it's considered Unknown. Note that here `p` is a predicate and `x` is a variable.

It is for this reason that we used the ProofWriter OWA dataset and removed the Unknowns; this gives us a dataset with reasoning traces concluding in either True or False. If we used CWA we would only have traces that could conclude True.

We evaluate the SI on 5 tasks from the ProofWriter dataset, each requiring varying numbers of reasoning steps (1, 2, 3 and 5). This requires the model to learn to compute intermediate conclusions that may not directly lead to the final answer, but may be needed to reach the final answer. While, this can be very hard to achieve using prompt engineering alone, we endeavour to do so, by demonstrating examples of intermediate and final steps of reasoning (for problems of depth >1). This means that the language model sees (1) examples that both encourage the model to select rules and facts that may not answer the problem in one-step but may help the model to obtain an intermediate output that can be used in a future step and (2) examples of the final step, which takes the model to the final answer. See Sec. C.2 for an example prompt.

The ProofWriter tasks involve predicting if a given statement, for example `"Bob is nice."`, is `True` or `False` given the context of facts and rules. Our SI model attempts to derive the statement `"Bob is nice."` or the negation of the statement `"Bob is not nice."` from the context. To assign a label `True` or `False` we follow the procedure proposed in the original ProofWriter paper (Tafjord et al., 2021) and test if any of the implications matches the given statement. If there is a match, the statement is considered to be True, otherwise False.

ProofWriter results in Fig. 4b show that the Selection-Inference model outperforms the baselines for problems of depth zero and one, however, with increasing depth, the gap between SI and the baselines diminishes. This is in part because prompt engineering is not sufficient to obtain an optimal Selection module.

Another challenge with the ProofWriter dataset is deciding how many arguments should be selected for each rule. In the ProofWriter dataset, some rules take one argument, others take two. We experimented with various different ways to encourage the model to stop selecting arguments. For example, we append `". Therefore, "` as a choice to the context that the model can select. If the language model selects `". Therefore, "` then the selection step ends. We allowed a maximum of two facts to be selected. Note that when using fine-tuned models, estimating the number of statements via heuristics is no longer necessary since the model learns to generate a string with the required number of sentences referenced by sentence label. For example, "sent 1. We know that sent 7 and sent 5."

To obtain results in Fig. 4b we run SI model for the minimum number of steps needed to solve the problem; a Depth $d$ problem is run for $d$ steps, with the exception of the depth-0 problem which is run for 1 step. However, models may perform better when allowed to run for additional steps, in the case where the model makes a mistake on one step, but later recovers. Fig. A1 shows how the number of SI steps can lead to improved performance. There is greater improvement to performance for depth-1 reasoning. For depth-2 reasoning, there was not enough variation in the selections at each step, so additional steps did not help as much as for depth-1. If the model is run for additional steps, once the answer has been reached, then the model often repeats its final reasoning step for the remainder of the iterations.

# E   REASONING TRACES OUTPUT BY SI

## E.1   BABI 15

Below we show examples of reasoning traces output by our SI framework for the bAbI 15, deductive reasoning task. These examples are ones that the model got correct but are otherwise not cherry picked for their reasoning quality.

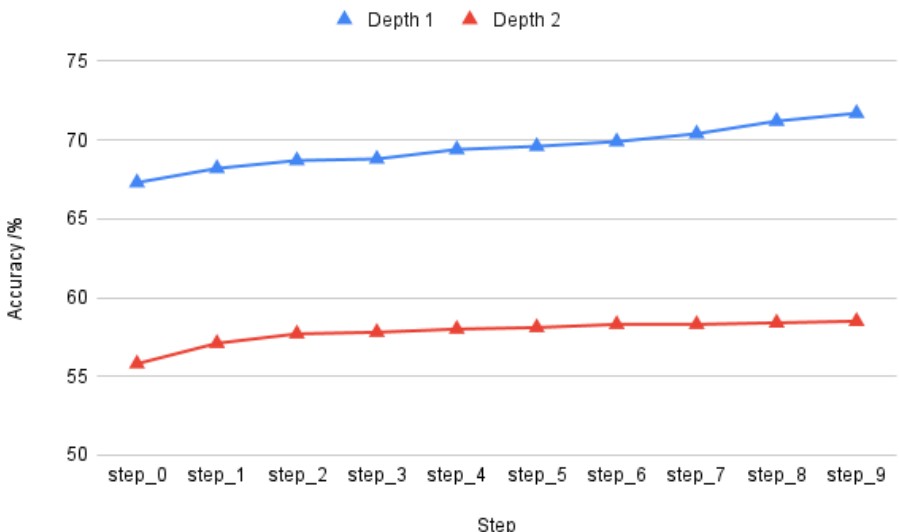

Figure A1: **ProofWriter: effect of additional reasoning steps.** With additional iterations of selection and inference the probability of the model producing the correct answer increases.

**bAbI 15 Example 1**

```
wolves are afraid of mice
sheep are afraid of mice
winona is a sheep
mice are afraid of cats
cats are afraid of wolves
jessica is a mouse
emily is a cat
gertrude is a wolf
Question: what is emily afraid of?

Selection: emily is a cat. We know that cats are afraid of wolves
Inference: emily is afraid of wolves
```

**bAbI 15 Example 2**

```
wolves are afraid of mice
sheep are afraid of mice
winona is a sheep
mice are afraid of cats
cats are afraid of wolves
jessica is a mouse
emily is a cat
gertrude is a wolf
Question: what is winona afraid of?

Selection: winona is a sheep. We know that sheep are afraid of mice
Inference: winona is afraid of mice
```

**bAbI 15 Example 3**

```
wolves are afraid of mice
sheep are afraid of mice
winona is a sheep
```

```
mice are afraid of cats
cats are afraid of wolves
jessica is a mouse
emily is a cat
gertrude is a wolf
Question: what is gertrude afraid of?

Selection: gertrude is a wolf. We know that wolves are afraid of mice
Inference: gertrude is afraid of mice
```

**bAbI 15 Example 4**

```
wolves are afraid of mice
sheep are afraid of mice
winona is a sheep
mice are afraid of cats
cats are afraid of wolves
jessica is a mouse
emily is a cat
gertrude is a wolf
Question: what is jessica afraid of?

Selection: jessica is a mouse. We know that mice are afraid of cats
Inference: jessica is afraid of cats
```

E.2   BABI 16

In the example below there are two possible conclusions that may be drawn. Since `brian`, `bernard` and `greg` are all rhions but brian and bernard are white, while greg is green the answer could be green or white. A black-box model may select one or the other answer without explaining why. Our model on the other hand shows that it use the facts that `greg is green` to infer that `julius is green`.

**bAbI 16 Example 1: Multiple "correct" answers**

```
bernhard is a rhino
brian is a rhino
bernhard is white
brian is white
lily is a lion
lily is yellow
greg is a rhino
greg is green
julius is a rhino
Question: what color is julius?

Selection: julius is a rhino. We know that greg is a rhino and greg is green
Inference: julius is green
```

In the example below the model must answer the question `what color is greg?`. `greg` is a lion, which means the model needs to infer a rule about lions. However, on the first step of reasoning the model infers that `we inferred that swans are often gray`. While this is factually correct given the context, it does not help to answer the question. However, the model is able to recover and correctly infer that `we inferred that lions are often white` meaning that the model is still able to solve the problem despite making a mistake.

**bAbI 16 Example 2: Recovering from an error**

```
julius is a swan
julius is gray
bernhard is a swan
```

**Selection model training example:**

Prompt:

sent 0: If something is rough then it visits the lion.
sent 1: the cow visits the lion.
sent 2: the lion visits the cow.
sent 3: the cow is rough.
sent 4: If something visits the cow and the cow visits the lion then the lion is nice.
sent 5: If the something is nice it visits the lion.
Does it imply that the statement "The lion visits the lion" is True?
Target:
sent 4. We know that sent 2 and sent 1. Therefore, the lion is nice.

**Inference model training example:**

Prompt:

If something visits the cow and the cow visits the lion then the lion is nice. We know that the the lion visits the cow and the cow visit the lion.

Target
Therefore, the lion is nice.

Figure A2: **Examples of inputs and targets used to fine-tune LLM for the Selection and Inference modules.** On the is an example of supervision for step-1 in a depth-2 problem. We create training data for all steps in the dataset by appending the ground-truth inferences for the intermediate steps to the context. On the right, is an example of a data sample used to train the inference model. Both the Selection and Inference LLM's are fine-tuned on only single steps of reasoning.

```
bernhard is yellow
brian is a lion
greg is a lion
brian is white
lily is a frog
lily is gray
Question: what color is greg?

Selection: julius is a swan. We know that julius is gray
Inference: we inferred that swans are often gray

Selection: brian is a lion. We know that brian is white
Inference: we inferred that lions are often white
```

## F  FINE-TUNING SELECTION-INFERENCE DETAILS AND EXTRA RESULTS

Fig. A2 shows examples of the format of the data used to fine-tune the Selection and Inference LLMs. The Selection module is trained to predict sentence labels rather than the sentence strings. This prevents the model from making up facts and forces the model to use information in the context.

During selection, rather than scoring each sentence we can append sentence labels to each element in the context (as shown in Fig. A2); use the Selection LLM to generate selection strings and substitute in elements from the context using a dictionary. This process is much faster than scoring each element of the context, but still ensures that the selection consists only of samples from the context; the Selection module cannot make up facts to answer the question. Fig. A3 shows examples of entailment computed by the Inference LLM after fine-tuning.

Fig. 4b compares SI models incorporating fine-tuned vs. prompt-engineered LLMs. We see that the model using LLMs fine-tuned on single steps of reasoning significantly outperform both the prompt-engineered LLMs and a Vanilla LLM prompt-engineered to predict the final answer directly. Fig. A4 shows a reasoning trace output by the SI model on a challenging depth-5 problem.

### F.1  SELECTION-INFERENCE REASONING TRACES

### F.1.1  DEPTH-2 REASONING TRACES FOR DEPTH-2 PROBLEMS

Below we show examples of depth-2 reasoning traces produced via Selection-Inference using modules fine-tuned on the ProofWriter dataset.

**Example 1:**

```
If someone is cold then they eat the lion.
If someone is blue and they chase the dog then the dog chases the lion.
```

| candidate | prompt |
|---|---|
| Therefore, Harry is big | If something is furry then it is big. We know that Harry is furry. |
| Therefore, Anne is not cold | If Anne is young and Anne is smart then Anne is not cold. We know that Anne is young and Anne is smart. |
| Therefore, Harry is nice | If someone is white then they are nice. We know that Harry is white. |
| Therefore, the bald eagle eats the squirrel | If someone chases the cat then they eat the squirrel. We know that the bald eagle chases the cat. |
| Therefore, the cat is big | All round, rough people are big. We know that the cat is round and the cat is rough. |
| Therefore, Dave is blue | All green, young things are blue. We know that Dave is green and Dave is young. |
| Therefore, Erin is green | If someone is red then they are green. We know that Erin is red. |
| Therefore, Anne is rough | All big things are rough. We know that Anne is big. |
| Therefore, Harry is smart | Green, cold people are smart. We know that Harry is green and Harry is cold. |
| Therefore, Bob is young | Big people are young. We know that Bob is big. |
| Therefore, the dog visits the dog | If something is blue then it visits the dog. We know that the dog is blue. |
| Therefore, Dave is round | If someone is kind and furry then they are round. We know that Dave is kind and Dave is furry. |
| Therefore, the mouse is green | If the mouse visits the tiger then the mouse is green. We know that the mouse visits the tiger. |

Figure A3: **Inference Module fine-tuning test examples**

**Problem:**
If something is quiet and cold then it is smart.
All young, cold things are kind.
If something is kind then it is nice.
All smart, blue things are kind.
If something is cold then it is quiet.
If Anne is smart and Anne is kind then Anne is cold.
If something is nice and blue then it is young.
If something is nice and young then it is cold.
Anne is young.
Gary is nice.
Charlie is nice.
Charlie is quiet.
Anne is blue.
...(removed some facts for brevity)...
Dave is smart.
Anne is quiet.
Dave is blue.
Gary is cold.
Anne is nice.
Does it imply that the statement "Dave is not quiet" is True?

**Reasoning trace output by the model:**
**Selection**: All smart, blue things are kind. We know that Dave is smart and Dave is blue.
**Inference**: Dave is kind.

**Selection**: If something is kind then it is nice. We know that Dave is kind.
**Inference**: Dave is nice.

**Selection**: If something is nice and blue then it is young We know that Dave is nice and Dave is blue.
**Inference**: Dave is young.

**Selection**: If something is nice and young then it is cold We know that Dave is nice and Dave is young.
**Inference**: Dave is cold.

**Selection**: If something is cold then it is quiet. We know that Dave is cold.
**Inference**: Dave is quiet.

Figure A4: A ProofWriter depth-5 reasoning trace output by our model. The model produces an interpretable reasoning trace that allows us to inspect how the model reached its answer.

```
If someone eats the dog then the dog is young.
If someone is young and they eat the lion then they are red.
If someone is nice then they eat the dog.
If someone
    eats the lion and the lion eats the dog then the dog eats the lion.
If someone
    sees the lion and the lion chases the dog then the lion is nice.
If the lion sees the dog and the dog sees the lion then the dog is nice.
the lion sees the dog.
the dog sees the lion.
Does it imply that the statement "The dog eats the dog" is True?
step 0:
```

```
Selection
    :  If the lion sees the dog and the dog sees the lion then the dog is
    nice. We know that the lion sees the dog and the dog sees the lion.
Inference: The dog is nice.
step 1:
Selection:  If
    someone is nice then they eat the dog. We know that The dog is nice.
Inference: The dog eats the dog.
```

**Example 2:**

```
If something is cold and red then it likes the mouse.
If something needs the cat then the cat sees the dog.
If something needs the cow then the cow sees the mouse.
If something sees the dog then the dog likes the cat.
If the cat is not green then the cat does not see the mouse.
If something sees the mouse then it is cold.
If something
    likes the cat and the cat needs the cow then the cow is nice.
If something sees the cow then it needs the cow.
the mouse needs the dog.
the mouse needs the cat.
the dog is nice.
the cat is green.
the mouse is not nice.
the dog needs the cat.
the dog sees the cow.
the cow is not red.
the cat likes the dog.
the mouse sees the cow.
the mouse needs the cow.
the cow sees the dog.
the mouse is green.
the cow needs the dog.
the mouse is blue.
Does it imply that the statement "The cow is cold" is True?
step 0:
Selection:  If something needs the cow
    then the cow sees the mouse. We know that the mouse needs the cow.
Inference: The cow sees the mouse.
step 1:
Selection:  If something
    sees the mouse then it is cold. We know that The cow sees the mouse.
Inference: The cow is cold.
```

**Example 3:**

```
Rough things are white.
If Erin is smart and Erin is rough then Erin is white.
If something is round then it is rough.
If Erin is round and Erin is not smart then Erin is white.
All quiet things are not white.
If something is blue and white then it is not quiet.
Erin is green.
Erin is rough.
Erin is blue.
Does it imply that the statement "Erin is not quiet" is True?
step 0:
Selection:  If Erin is smart and Erin is rough
    then Erin is white. We know that Erin is green and Erin is rough.
Inference: Erin is white.
step 1:
Selection:  If something is blue and white
    then it is not quiet. We know that Erin is blue and Erin is white.
Inference: Erin is not quiet.
```

**Example 4:**

```
If something chases the squirrel then the squirrel is big.
If something is big then it is not kind.
If something chases the bald
    eagle and it sees the bald eagle then the bald eagle sees the lion.
the cow does not like the squirrel.
the cow sees the lion.
the bald eagle likes the lion.
the cow chases the squirrel.
the lion chases the cow.
the bald eagle is not round.
the squirrel likes the cow.
the cow likes the lion.
the cow chases the bald eagle.
the squirrel likes the bald eagle.
the cow is kind.
the lion chases the squirrel.
the cow does not see the squirrel.
the lion chases the bald eagle.
the squirrel likes the lion.
Does it imply that the statement "The squirrel is kind" is True?
step 0:
Selection:  If something chases the squirrel
    then the squirrel is big. We know that the cow chases the squirrel.
Inference: The squirrel is big.
step 1:
Selection:  If something
    is big then it is not kind. We know that The squirrel is big.
Inference: The squirrel is not kind.
```

F.1.2   DEPTH-3 REASONING TRACES FOR DEPTH-3 PROBLEMS

Below we show examples of depth-3 reasoning traces produced via Selection-Inference using modules fine-tuned on the ProofWriter dataset.

**Example 1:**

```
If something eats the bald eagle then it eats the squirrel.
If something
    eats the squirrel then the squirrel does not eat the bald eagle.
If the bald eagle is not red then the bald eagle likes the squirrel.
If the squirrel
    likes the bald eagle then the squirrel visits the bald eagle.
If something likes the bald eagle then the bald eagle is young.
If something is young then it eats the squirrel.
the squirrel visits the bald eagle.
the bald eagle visits the squirrel.
the squirrel likes the bald eagle.
Does it imply that
    the statement "The squirrel does not eat the bald eagle" is True?
step 0:
Selection:  If something likes the bald eagle then the
    bald eagle is young. We know that the squirrel likes the bald eagle.
Inference: The bald eagle is young.
step 1:
Selection:  If something is young
    then it eats the squirrel. We know that The bald eagle is young.
Inference: The bald eagle eats the squirrel.
step 2:
Selection:  If something eats the squirrel then the squirrel does not
    eat the bald eagle. We know that The bald eagle eats the squirrel.
Inference: The squirrel does not eat the bald eagle.
```

**Example 2:**

```
    If someone is quiet then they are white.
If someone is young and red then they are white.
Young people are nice.
If someone is nice then they are round.
All quiet people are young.
Red, big people are nice.
Round, red people are white.
If someone is round then they are quiet.
Harry is young.
Bob is red.
Bob is big.
Does it imply that the statement "Harry is not quiet" is True?
step 0:
Selection:  Young people are nice. We know that Harry is young.
Inference: Harry is nice.
step 1:
Selection:
     If someone is nice then they are round. We know that Harry is nice.
Inference: Harry is round.
step 2:
Selection:
    If someone is round then they are quiet. We know that Harry is round.
Inference: Harry is quiet.
```

**Example 3:**

Here the model make the same selection on `step 1` and `step 2` and therefore does not correctly show that the "The lion visits the mouse". Although the model has failed here it is (a) possible that the model can still recover (if ran for additional steps) and compute the correct answer and (b) we can see how the model reached its answer and interpret its failings because the reasoning trace is causal.

```
If something is green then it visits the mouse.
If something chases the bear then it is green.
If something chases the mouse then the mouse sees the bear.
If something sees the
     bear and the bear visits the mouse then the mouse chases the lion.
If something chases the mouse then it is green.
If something visits the bear then it chases the mouse.
the mouse visits the lion.
the lion visits the bear.
the bear chases the lion.
Does it
    imply that the statement "The lion does not visit the mouse" is True?
step 0:
Selection:  If something visits the
    bear then it chases the mouse. We know that the lion visits the bear.
Inference: The lion chases the mouse.
step 1:
Selection:  If something chases
     the mouse then it is green. We know that The lion chases the mouse.
Inference: The lion is green.
step 2:
Selection:  If something chases
    the mouse then it is green. We know that The lion chases the mouse.
Inference: The lion is green.
```

## G  LIMITATION DETAILS

We have seen that our approach to solving reasoning problems, using SI, has many desirable properties and indeed this model is intended to be a proof of concept to demonstrate that it is possible to build a model with these properties. However, as a proof of concept, this model has several limitations, which we now discuss in detail.

When observing the outputs of our model, the main point of failure tends to be the Selection module. This is hard to quantify since we do not have labelled data for the intermediate reasoning steps. One reason for this is that we use prompt-engineering to encourage language models to produce the correct outputs, rather than fine-tuning. While our current results are good, and do not require (large amounts of) task specific data, they can be significantly improved by fine-tuning our models for specific tasks, as demonstrated in Section 6.

Prompt engineering works best for single modality cases (Nakano et al., 2021). It is more difficult to get the model to do multi-step reasoning since the distribution or patterns for the intermediate steps differ to the final step. It is also difficult to get the model to figure out how many arguments to select or how many arguments a rule takes, using only prompt engineering, again because there are multiple different patterns that the model needs to learn how and when to apply.

Other limitations of our work include the assumption that a database of facts or rules are given. In many practical settings we would need to be able to retrieve relevant knowledge from an existing knowledge base. There is exciting progress being made in this area (Lazaridou et al., 2022) and we hope in the future to combine these approaches with the SI model.

Finally, while our model has the benefit that performance scales with compute time; the longer we run our model the more likely it is to reach a correct answer, we do not have a good way of deciding when to halt the reasoning process or to filter the reasoning traces. In our current approach we have fixed budget, and tend to report results based on the final inference. Results in Fig. A1 suggests that accuracy could be improved if we had a mechanism for filtering the reasoning steps and selecting the best answer.

## H    BASELINE DATASETS

In this paper we used tasks from six sources: bAbi (Weston et al., 2016), BigBench (Ghazal et al., 2017), AAC (Betz et al., 2021), Jeopardy (Tunguz, 2019), ProofWriter (Tafjord et al., 2021) and 2WikiMultiHop (Ho et al., 2020). All of these dataset are publicly available and our use of the data was in accordance to their respective license permissions. As far as we are aware, none of the datasets contain personally identifiable information or offensive content.

**Task decomposition**    From bAbI dataset (Weston et al., 2016) used tasks 1-3 to measure the ability of LLMs to cope with progressively larger numbers of reasoning steps on the same type of problem; task 6 to compare to task 1 whether yes/no questions are easier for the models to answer compared to more free-form answers; task 9 to compare to task 1 to test how well the models can deal with negation; task 10 to test whether LLMs know that the facts they are given are not sufficient to answer a question; tasks 15 and 16 to test basic deduction and induction abilities respectively; task 18 to compare to task 2 for two step reasoning on a different kind of task (based on size).

Jeopardy (Tunguz, 2019) and 2WikiMultiHop (Ho et al., 2020) tasks measure the ability of LLMs to do reasoning in less structured settings, where the answer has to be generated in free form, and the level of available context varies between none (2WikiMultiHop, Jeopardy), to relevant and irrelevant unstructured context (2WikiMultiHop With Context and 2WikiMultiHop With Context & Facts), to relevant structured context only (2WikiMultiHop With Evidence).

AAC (Betz et al., 2021) measures the ability of LLMs to do relatively shallow formal reasoning (1-2 steps) over a relatively large set of syllogistic argument schemes, both with (AAC Split Extended) and without (AAC Split) dealing with negation.

ProofWriter (Tafjord et al., 2021) tasks measure the ability of models to do formal reasoning over a progressively more difficult tasks that require more steps of reasoning.

From BigBench (Ghazal et al., 2017) we imported the following tasks: Analytic Entailment, Epistemic Reasoning and measure the ability of LLMs to decide implicit entailment relationship given facts.

Entailed Polarity, Presuppositions as NLI and Logical Arguments measure the ability of LLMs to understand implied information from vague language.

Evaluating Information Essentiality and Sufficient Information measure how well LLMs can evaluate which context information is relevant and sufficient to answer a question.

Formal Fallacies Syllogisms Negation, Logic Grid Puzzle, Logical Fallacy Detection, and Logical Deduction test the ability of LLMs to do formal deductive reasoning.

Sequence Problems Tasks and Tracking Shuffled Objects are similar to bAbI tasks 1-3 and evaluates the ability of LLMs to do multi-hop reasoning based on sequenced facts.

Physics Questions and Unit Interpretation measure the ability of LLMs to reason about grade school science problems.

StrategyQA measures the ability of LLMs to do multi-hop reasoning based on general knowledge that is not explicitly provided as context.

**Multiple choice normalisation**    We evaluated whether normalising log probability of the choices under the model by the token length resulted in better accuracy to avoid potential bias as reported in Lin et al. (2020). We found significant ($p\!=\!0.0002$, two-tailed t-test with equal variance) but minimal difference between the average accuracy across all evaluated tasks, when evaluated with ($67.92\!\pm\!46.68\%$) and without normalisation ($68.3\!\pm\!46.53\%$). For this reason we use the unnormalised measures in the paper.

**Evaluating dataset bias**    To check whether the datasets we use are biased, we compared how much the baseline multiple choice accuracy of LLMs when presented with options all by themselves, without any context or question, deviates from the expected random performance calculated as $1/N$, where $N$ is the number of choices. We found that the two differed by a very small amount $0.08\!\pm\!8.74\%$ on average across all models and all multiple-choice datasets ($p\!=\!5e\!-\!16$, two-tailed t-test with equal variance). Since the effect size was so small, we concluded that the datasets were not biased and present the expected random baseline where appropriate.

**Appending choices to bAbI tasks**    We evaluated whether adding choices to the prompt improved the multiple choice accuracy of LLMs on bAbI tasks. We found that this was not the case, with the average accuracy across all bAbI tasks being $37.86\!\pm\!48.5\%$ when choices were appended, and $44.86\!\pm\!49.74\%$ when choices were not appended ($p\!=\!2e\!-\!61$, two-tailed t-test with equal variance). For this reason, we report the latter results in the paper.

**2WikiMultiHop results**    We evaluated the performance of LLMs on 2WikiMultiHop (Ho et al., 2020) dev subset using exact string match between the generated and the ground truth answers. In particular, the generated answer was truncated at the first sentence up to ".", "?", "!", ";" or newline characters following the BigBench generative evaluation protocol (Ghazal et al., 2017). The two answers were then both stripped of all punctuation, white space and special characters before comparison is made. Dataset examples receive a score of 1 if the post-processed answers match exactly, and 0 otherwise.

We found that the models scored $13.62\!\pm\!34.3\%$ on average on the original dataset, consisting of questions only. When the context of Wikipedia paragraphs with relevant and irrelevant facts to answer the question was added to the context, the performance dropped to $1.47\!\pm\!12.02\%$ on average. Adding information about the relevant facts within these context paragraphs did not help much, resulting in $1.97\!\pm\!13.9\%$ accuracy. On the other hand, adding only the relevant facts extracted from the underlying knowledge graph triples has more than doubled the models' performance, resulting in $35.55\!\pm\!47.87\%$ average accuracy.

**General insights**    LMs get progressively worse as more steps are needed for reasoning (see bAbI tasks 1-3 and 18, ProofWriter tasks; although not the case for Logical Deduction, Sequence Problems Tasks and Tracking Shuffled Objects is at chance).

LMs find yes/no questions harder to answer than freeform questions (see bAbI task 6 vs 1).

LMs struggle to deal with negation (bAbi task 9 vs 1) unless it is in a very well formalised limited setup (AAC Split Extended).

LMs struggle with deciding when they do not have sufficient information (bAbI task 10 vs 1; Evaluating Information Essentiality and Sufficient Information are both at chance).

LMs are average at formal deduction and induction tasks, although their performance very much depends on the difficulty of the task and the evaluation protocol (see bAbI task 15 and Logical Deduction, although Proof Writer, Formal Fallacies Syllogisms Negation, Logic Grid Puzzle, and

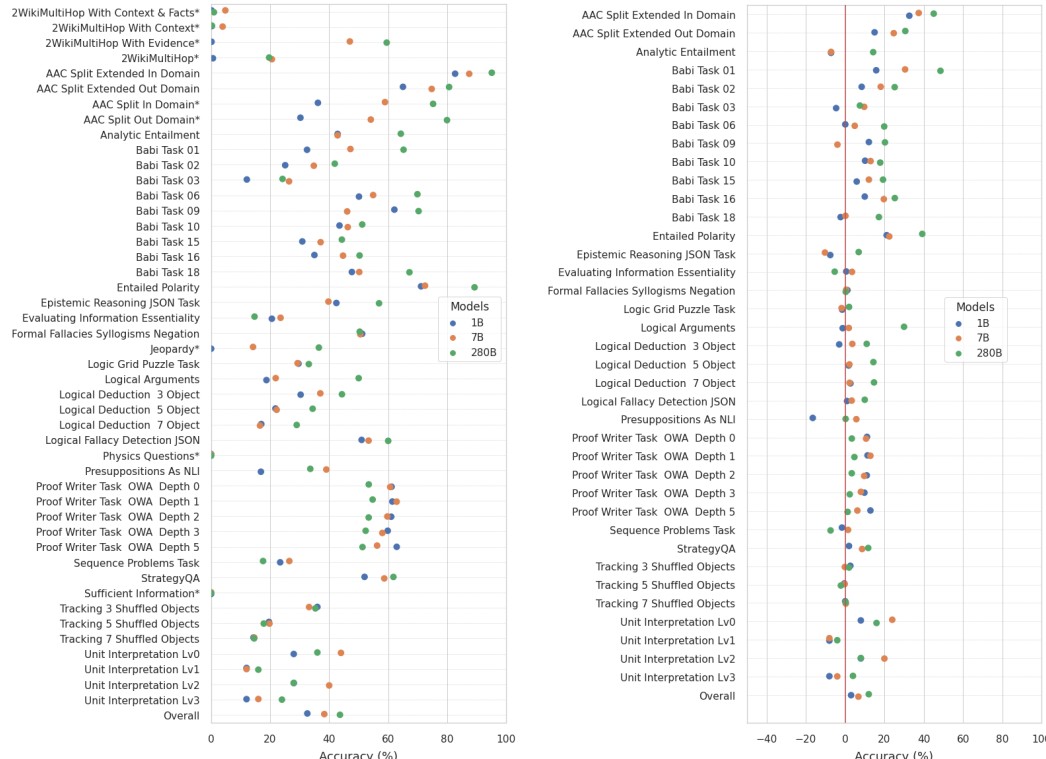

(a) Absolute multi-choice and generative (marked with an asterix *) accuracy.

(b) Relative accuracy compared to chance level for multi-choice tasks. Red line - chance performance.

Figure A5: Average accuracy of 1B, 7B and 280B parameter Vanilla LLMs evaluated on logical reasoning tasks in a 5-shot generalisation setting.

Logical Fallacy Detection results are close to chance, while AAC results, where the models are evaluated in a very structured setting are very good).

In less formal mutli-hop question answering scenarios, LLMs are close to chance when no context is provided (2WikiMultiHop, StrategyQA; although Jeopardy is an outlier) or when the provided information is unstructured (e.g. whole Wikipedia paragraphs as in 2WikiMultiHop With Facts and 2WikiMultiHop With Facts & Rules), but do better when minimal structured context information is provided (2WikiMultiHop With Evidence).

LMs also perform poorly in solving grade school science problems (Physics Questions and Unit Interpretation) although this ability is better in multiple choice compared to generative evaluation settings.

LMs are, however, reasonable at doing simple implication, entailment and induction tasks (see bAbI task 16, Analytic Entailment, Entailed Polarity, and Logical Arguments; although on Epistemic Reasoning and Presuppositions as NLI the models perform around chance level).

## I  TESTS OF STATISTICAL SIGNIFICANCE

To calculate statistical significance of differences between different models in Fig. 4 we used proportion hypothesis test for binary data. In particular, we used two-sided `proportions_ztest` from `statsmodels.stats.proportion`.

## J  CoT EVALUATION

A preliminary human evaluation test was run on 50 randomly sampled problems from each of the five bAbI tasks used in this paper, evaluating CoT and SI approaches based on three metrics:

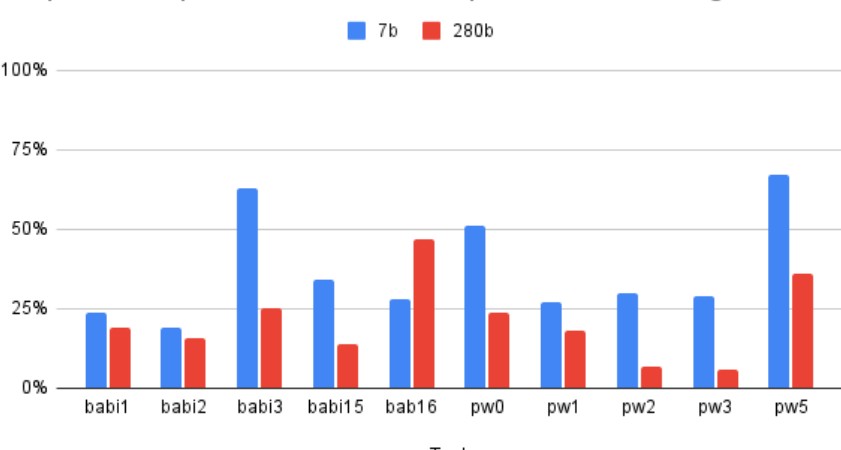

Figure A6: Proportion of problems in each dataset where COT generated a reasoning trace with made up facts.

1. Are the facts selected within the reasoning trace faithful to the facts presented in the context?
2. Even if the facts are made up, is the selection logically valid?
3. Even if the selection is irrelevant to the question, is the resulting inference logically consistent?

Full results are presented in Table A1. Some representative qualitative examples of the CoT reasoning are included below:

```
INPUT:
brian is a swan
greg is a lion
greg is white
brian is gray
lily is a rhino
lily is gray
julius is a frog
julius is white
bernhard is a swan
Question: what color is bernhard?
OUTPUT:  bernhard is a swan. We know that
    greg is a lion and that greg is white. Therefore, bernhard is white

INPUT:
greg is a frog
greg is white
bernhard is a swan
lily is a lion
brian is a rhino
bernhard is yellow
brian is green
lily is yellow
julius is a frog
Question: what color is julius?
OUTPUT:  julius is a frog. We know that
    brian is a frog and that brian is green. Therefore, julius is green
```

The above example shows CoT making up a fact to answer the question. The question above asks "what color is julius?". The CoT method correctly identifies that Julius is a frog. Then, rather than correctly saying "We know that greg is a frog" the model makes up a fact "brian is a frog" and combines this with the (existing) fact that "brian is green" to incorrectly conclude that "julius is green". Our

model, Selection-Inference, could not make this mistake, since the selection step is restricted to only selecting statements from the context (see Section 4.1).

```
INPUT:
lily is a frog
brian is a lion
greg is a rhino
julius is a rhino
julius is white
brian is green
lily is green
greg is white
bernhard is a frog
Question: what color is bernhard?
OUTPUT:  bernhard is a frog. We know that
    brian is a frog and that brian is green. Therefore, bernhard is green

INPUT:
cats are afraid of wolves
mice are afraid of cats
sheep are afraid of mice
gertrude is a cat
wolves are afraid of sheep
jessica is a mouse
emily is a wolf
winona is a cat
Question: what is emily afraid of?
OUTPUT:  emily is a cat. We know
    that cats are afraid of wolves. Therefore, emily is afraid of wolves.

INPUT:
at t=0 john travelled to the office
at t=1 mary journeyed to the kitchen
at t=2 mary moved to the garden
at t=3 daniel went to the office
at t=4 john went to the hallway
at t=5 mary moved to the kitchen
at t=6 mary travelled to the bedroom
at t=7 daniel journeyed to the garden
Question: where is mary?
OUTPUT:  at
    t=7 daniel journeyed to the garden. Therefore, mary is in the garden
```

## K  COMPARING CoT AND SI COMPUTATION

CoT involves a single forwards pass sampling `max_cot_tokens`. For a problem with H reasoning steps this would equate to roughly `cot_tokens_per_step = max_cot_tokens / H`.

Selection-Inference on the other hand does not involve a single forward pass, but rather (scores and) samples sequentially. For simplicity, let us initially assume that in the selection step we sample tokens rather than score them. This means that we sample `H * (max_selection_tokens + max_inference_tokens)`. So in the case where `cot_tokens_per_step = max_selection_tokens + max_inference_tokens` the number of computations is similar. This is a reasonable assumption since models were prompted to solve these problems in a similar way and we do not expect one model to solve problems with significantly more or less tokens than another model.

Now consider that in the selection step, we do not sample tokens auto-regressive (as in CoT) but rather we score tokens. Tokens can be scored in parallel and therefore can be computed more efficiently than they can be sampled. Therefore, if implemented correctly, Selection-Inference can be at least as computationally efficient as CoT, if not more efficient.

## L  HUMAN EVALUATION

The reasoning traces produced by CoT and SI approaches on the bAbI datasets were evaluated for correctness by two of the authors of this paper. For each dataset 50 random problems were evaluated. For each problem the raters were shown the context, question, correct answer and reasoning trace produced by the model (the reasoning trace included the final answer produced by the model). The raters were asked to answer the following questions:

1. Is the selection in the context: y/n
2. Is the selection logically valid: y/n
3. Does the inference follow from the selection: y/n

Note that for SI all selections are always in the context by construction. See Table A1 for the full results breakdown.

| Task | Selection in context | | Valid selection | | Inference correct | |
|---|---|---|---|---|---|---|
| | CoT | SI | CoT | SI | CoT | SI |
| bAbI 1 | 82% | 100% | 100% | 100% | 52% | 96% |
| bAbI 2 | 68% | 100% | 12% | 24% | 16% | 24% |
| bAbI 3 | 6% | 100% | 4% | 12% | 2% | 2% |
| bAbI 15 | 62% | 100% | 68% | 100% | 68% | 100% |
| bAbI 16 | 42% | 100% | 52% | 62% | 52% | 58% |

Table A1: Human evaluation of the quality of the reasoning traces produced by CoT and SI approaches on a subset of bAbI problems.

## M  SI ON ENTAILMENTBANK

This section contains a randomly sampled set of problems and their SI solutions from the EntailmentBank dataset.

### M.1  DEPTH 1

```
the mantle is a layer of the earth
the mantle is located just below the crust of the earth
What layer of Earth is located just below
    the crust? Inner core OR Mantle OR Continental shelf OR Outer core.
step 0:
Selection: the mantle is a layer of the earth.
    We know that the mantle is located just below the crust of the earth
Inference: the
    mantle is a layer of the earth that is located just below the crust
Correct Answer: mantle is the layer located below the crust of the earth

a toaster is made of a heating element for toasting
a heating element
    converts electrical energy / electricity into heat energy for heating
What energy change
    takes place when a piece of bread is toasted in a toaster? chemical
    energy to light energy OR electrical energy to heat energy OR
    heat energy to chemical energy OR light energy to electrical energy.
step 0:
Selection: a toaster is made
    of a heating element for toasting. We know that a heating element
    converts electrical energy / electricity into heat energy for heating
Inference:
    the toaster converts electrical energy into heat energy for toasting
Correct Answer
    : a toaster converts electrical energy to heat energy for toasting
```

```
a seat belt
     applies force in the opposite direction of the passenger 's motion
seat belts are used for preventing injuries to passengers in cars
In a car accident, a seat belt helps prevent injuries by applying
     a force less than the force of the moving passenger OR greater
     than the force of the moving car OR in the same direction as the
     car's motion OR in the opposite direction of the passenger's motion.
step 0:
Selection: a seat belt applies force
     in the opposite direction of the passenger 's motion. We know that
     seat belts are used for preventing injuries to passengers in cars
Inference
    : seat belts are used for preventing injuries to passengers in cars
     by applying force in the opposite direction of the passengers motion
Correct Answer: seat belts are used for preventing
     injuries by applying opposite force to the passenger's motion

metal is flexible
a wire is usually made of metal
Flexibility is a physical property of some matter. Which
     of these materials best demonstrates the property of flexibility
    ? a mirror OR a pencil OR a metal wire OR a telephone pole.
step 0:
Selection
    : metal is flexible. We know that a wire is usually made of metal
Inference: metal is flexible and a wire is made of metal
Correct Answer: a metal wire is flexible

a trunk is a part of a tree for supporting the tree
providing support is a kind of function
The main function
     of a tree's trunk is to provide air OR fruit OR sunlight OR support.
step 0:
Selection: a trunk is a part of a tree for supporting
     the tree. We know that providing support is a kind of function
Inference: the function of a trunk is to support a tree
Correct Answer: a function of a tree's trunk is to provide support

gravity causes orbits
planets in the solar system orbit the sun
The force necessary to keep planets in orbit
     around the Sun is gravity. OR friction. OR magnetism. OR nuclear..
step 0:
Selection: gravity
    causes orbits. We know that planets in the solar system orbit the sun
Inference
    : gravity causes the planets in the solar system to orbit the sun
Correct
     Answer: gravity causes planets in the solar system to orbit the sun

iron is always magnetic
iron nails are made of iron
Which object most likely has magnetic properties
    ? iron nail OR plastic clip OR rubber eraser OR wooden spoon.
step 0:
Selection
    : iron is always magnetic. We know that iron nails are made of iron
Inference: iron nails are magnetic
Correct Answer: iron nails are magnetic
```

## M.2 DEPTH 2

```
the sun is a kind of star
the earth revolves around the sun
```

```
a complete revolution
    / orbit of a planet around its star takes 1 / one planetary year
earth is a kind of planet
About how long does it take Earth to make
    one revolution around the Sun? a day OR a week OR a month OR a year.
step 0:
Selection: the sun is a kind of star. We know that a complete revolution
    / orbit of a planet around its star takes 1 / one planetary year
Inference: a
    complete revolution of the sun around the earth takes one solar year
step 1:
Selection: a complete revolution of the sun around the earth
    takes one solar year. We know that the earth revolves around the sun
Inference: a
    complete revolution of the earth around the sun takes one solar year
Correct Answer: a complete
    revolution of the earth around the sun will take one earth year

as distance from
    an object decreases , the pull of gravity on that object increases
the moon is the celestial object that is closer to the earth than the sun
the gravitational pull of the sun on earth 's oceans causes the tides
the gravitational pull of the moon on earth 's oceans causes the tides
The influence of the Moon on the tides on Earth
    is greater than that of the Sun. Which best explains this? The Moon
    is more similar in composition to Earth than the Sun. OR The Moon
    has a smaller mass than Earth and the Sun. OR The Moon has a higher
    density than the Sun. OR The Moon is closer to Earth than the Sun..
step 0:
Selection: the moon is the celestial object that
    is closer to the earth than the sun. We know that the gravitational
    pull of the moon on earth 's oceans causes the tides and
   the gravitational pull of the sun on earth 's oceans causes the tides
Inference: the gravitational pull of the moon on earths
    oceans is closer to the earth than the gravitational pull of the sun
step 1:
Selection: the gravitational pull
    of the moon on earth 's oceans causes the tides. We know that the
   moon is the celestial object that is closer to the earth than the sun
Inference
    : the moon is the celestial object that causes the tides on earth
Correct Answer: the gravitational pull of the moon on
    earth affects the tides more than the gravitational pull of the sun

moons orbit planets
gravity causes orbits
the moon is a kind of moon
earth is a kind of planet
What keeps the Moon orbiting Earth? the Sun's gravity
    OR the Moon's rotation OR Earth's rotation OR Earth's gravity.
step 0:
Selection: moons orbit planets. We know that earth is a kind of planet
Inference: earths moons orbit earth
step 1:
Selection: gravity causes orbits. We know that earths moons orbit earth
Inference: gravity causes the moon to orbit around earth
Correct Answer: gravity causes the moon to orbit the earth

a tool is a kind of invention
a compass is a kind of tool
natural magnetism is used for pointing north by a compass
Which invention uses
    natural magnetism? satellite OR telescope OR compass OR binoculars.
step 0:
```

```
Selection: a compass is a kind of tool. We
    know that natural magnetism is used for pointing north by a compass
Inference: a compass
    is a kind of tool used for pointing north by natural magnetism
step 1:
Selection: a compass is a kind of tool used for pointing north
    by natural magnetism. We know that a tool is a kind of invention
Inference: a compass
    is a kind of invention used for pointing north by natural magnetism
Correct
    Answer: a compass is a kind of invention that uses natural magnetism

adaptations are used for survival
an animal can survive in an environment with little food by storing fat
a camel's humps are used for storing fat
Camels have humps on their backs
    that store fat, allowing them to survive for many days without food.
    This makes camels well suited to desert life. This characteristic is
    an example of adaptation. OR instinct. OR migration. OR hibernation..
step 0:
Selection: adaptations are used for survival. We know that an
    animal can survive in an environment with little food by storing fat
Inference: an
    example of an adaptation is storing fat for times when food is scarce
step 1:
Selection: an example of an adaptation is storing fat for times when
    food is scarce. We know that a camel's humps are used for storing fat
Inference: a camels
     humps are a kind of adaptation for storing fat when food is scarce
Correct Answer: an example of an adaptation is camel humps storing fat

sunlight is a kind of solar radiation
ultraviolet light causes sunburn
sunlight contains ultraviolet light
Which form of solar radiation causes
    sunburn? Visible OR Ultraviolet OR Infrared OR X-rays OR Radio waves.
step 0:
Selection: sunlight is a kind
     of solar radiation. We know that ultraviolet light causes sunburn
Inference
    : ultraviolet light is a kind of solar radiation that causes sunburn
step 1:
Selection: ultraviolet light is a kind of solar radiation that
    causes sunburn. We know that sunlight is a kind of solar radiation
Inference: ultraviolet light is a kind of sunlight that causes sunburn
Correct Answer: the ultraviolet in solar radiation causes sunburn

an automobile is usually a heavy object
a kilogram is a kind of unit for measuring mass
kilogram is used to measure heavy objects
Which is the most appropriate SI unit to use for expressing the mass
    of an automobile? millimeter OR milligram OR kilometer OR kilogram.
step 0:
Selection: an automobile is usually a
    heavy object. We know that kilogram is used to measure heavy objects
Inference: kilogram can be used to measure the mass of an automobile
step 1:
Selection: kilogram can be used to measure the mass of an automobile
    . We know that a kilogram is a kind of unit for measuring mass
Inference: kilogram can be used to measure the mass of an automobile
Correct Answer: kilogram can be used to measure the mass of an automobile
```

