# OpenReview forum: "Selection-Inference: Exploiting Large Language Models for Interpretable Logical Reasoning"
_ICLR.cc/2023/Conference — ICLR 2023 notable top 5%_

### Official Review · Reviewer_BSuH · 2022-10-17

**Confidence:** 4
**Correctness:** 3
**Technical Novelty And Significance:** 3
**Empirical Novelty And Significance:** 4
**Recommendation:** 8

**Clarity, Quality, Novelty And Reproducibility:**

The paper somewhat novel, well written and easy to read, however some things could be more clear, see below for clarification suggestions:

**Clarity (c1)** : The paper never mentions what type of model is used. The authors claim it is to preserve anonymity, but once a model is public, anyone can use it given the available compute resources. Saying that a 280B model was used is also having an impact on the anonymity of the work but it is still valuable information, just like the model type. At minimum the authors should discuss if the model is an encoder-decoder or decoder-only architecture.

**Clarity (c2)** The abstract of the paper initially claims to evaluate models on a suite of 46 tasks, however the SI framework in the main result sections is evaluated on Baby (Sec.5) and ProofWriter (Sec.5 & Sec.6) only. This makes the paper misleading. It may be better to announce ProofWriter and Baby in the abstract as this seems to be the main focus.
If that is not the main focus, then why are the 46 tasks in the Appendix (Figure A5)? It is not clear if the SI framework was used in these appendix tasks or some other type of prompting? More details on the division of tasks and what prompting mechanism has been used would be beneficial.

**Clarity (c3)** : The paper would be even more clear if it explicitly explained what is meant by “testing in 5-shot setting”. For instance does it ever happen that the 5 examples in the prompt, when tokenized, are longer than the input size supported by the model? What happens in this case?

Similarly, the paper should explain how the prompt examples were picked?

In the case of ProofWriter D0-D5, were the 5 examples in the model’s prompt from the same depth as the one being tested? In the ProofWriter paper, the authors had experiments where they trained on D3 and tested on D5 for instance, does this work explore compositional generalization like this as well with the prompts? If not, I think it would be an interesting result to test: prompting with D3 examples and testing on D5 examples.


**Strength And Weaknesses:**

**Strengths** : This is a good paper with good motivation. The work can have a positive impact on the field of multi-step reasoning. However some aspects are not entirely clear (more details in the next section). Some improvement suggestions are listed below:

**Weakness** : The Selection algorithm relies on the manual selection of K: number of facts to include in the prompt. This makes the framework hard to use in practice as manually selecting the number of required facts for each Selection step can be very tedious. Did the authors explore setting this parameter to something large (say 4 for all steps, for all tasks)?

Similarly and most importantly for the entire process of Selection & Inference the steps are repeated until a predefined number of steps. Hence the model is not responsible for identifying when it has answered the question and when it needs to stop this process.
Do the authors have some thoughts about how to resolve this issue in future work? It would be nice to discuss this in the final section of the paper.

**Question** : Do the authors have some intuition why in Figure 6b the baseline model performs better as the depth increases? The opposite would be a more natural behavior (like in the case of the SI model).

**Recommendation** : Although to a smaller scale, and with models trained from scratch, the idea of using language models to generate step by step reasoning chains was first explored in “Measuring systematic generalization in neural proof generation with transformers” (Gontier et. al, 2020). The paper should compare their approach to this one in the Related Work section.
Similarly, the authors could consider testing their Selection & Inference framework on the CLUTRR benchmark (Sinha et. al, 2019) in addition to Baby & ProofWriter.


**Summary Of The Paper:**

This work presents a new way to perform multi-step logical reasoning tasks with frozen pre-trained large language models. In particular they first select a few (most often two?) pieces of information (Selection) by scoring each fact in the context with the model’s log-likelihood. Then the retrieved facts are combined and the model generates a conclusion based only on those facts (Inference).

Results on ProofWriter and Baby tasks show that the Selection-Inference process is better than other traditional prompting methods (CoT). The work also shows improvement over models with more parameters.


**Summary Of The Review:**

Overall, this is a good paper with impactful research, however it would benefit some clarifications and a discussion on how to resolve the weakness mentioned above.

---

> ### Author Response · Authors · 2022-11-18
> **Thank you for reviewing our paper.**
>
> Thank you for reviewing our paper. Please see our response below.
>
> > Clarity (c1) :  At minimum the authors should discuss if the model is an encoder-decoder or decoder-only architecture.
>
> Thanks for this suggestion. The model is a decoder-only architecture, we have updated the paper in several places, including the abstract, to reflect this.
>
> > Clarity (c2) The abstract of the paper initially claims to evaluate models on a suite of 46 tasks, however the SI framework in the main result sections is evaluated on Baby (Sec.5) and ProofWriter (Sec.5 & Sec.6) only. This makes the paper misleading. It may be better to announce ProofWriter and Baby in the abstract as this seems to be the main focus.
>
> Thank you for pointing this out, we have updated the abstract in the revised paper as follow to reflect this:
>
> Revised Abstract: "​​Focusing on a sub-set of 10 reasoning tasks from ProofWriter and bAbI, we show that a 7B parameter, decoder-only LLM used within the SI framework in a 5-shot generalisation setting, with no fine-tuning, yields a performance improvement of over 100% compared to an equivalent vanilla baseline."
>
> > If that is not the main focus, then why are the 46 tasks in the Appendix (Figure A5)? It is not clear if the SI framework was used in these appendix tasks or some other type of prompting? More details on the division of tasks and what prompting mechanism has been used would be beneficial.
>
> The results in Figure A5 are a fine-grain breakdown of the results summarized in Figure 3 (Section 3, “How Well Do Large Language Models Reason?”, of the main text), showing Vanilla 5-shot language model performance. In the revised paper, we have added the following clarification to the Figure 3 caption:
>
> “A detailed per-task breakdown is shown in Figure A5.”
>
> We have also updated the caption in Figure 5A as follows:
>
> “Average accuracy of 1B, 7B and 280B parameter Vanilla LLMs evaluated on logical reasoning tasks in a 5-shot generalization setting.”
>
> > Clarity (c3) : The paper would be even more clear if it explicitly explained what is meant by “testing in 5-shot setting”.
>
> We follow the same protocol as that used for the BigBench evaluation in Rae et al. (2020). For each sample, this involves drawing random (question, answer) pairs from the rest of a well shuffled batch and showing these as examples in the prompt. Examples of prompts used for the Vanilla LLM evaluation are shown in Appendix A.
>
> We have updated Appendix A in the revised paper to include the following:
>
> “For each sample, a 5-shot prompt is composed by sampling five (question, answers) pairs from the batch and composing them according to a simple template. See examples below.”
>
> We have updated the text in Section 3 as follows:
>
> “To this end, we evaluated decoder-only LLMs of various sizes in a 5-shot setting, following the same protocol used for the BigBench evaluation in Rae et al (2020), on a larger set of 46 tasks.”
>
> > In the case of ProofWriter D0-D5, were the 5 examples in the model’s prompt from the same depth as the one being tested? In the ProofWriter paper, the authors had experiments where they trained on D3 and tested on D5 for instance, does this work explore compositional generalization like this as well with the prompts? If not, I think it would be an interesting result to test: prompting with D3 examples and testing on D5 examples.
>
> In Section 3, where we evaluate Vanilla LLM reasoning performance no reasoning traces are shown in the prompt.
>
> For our Chain-of-Thought baselines, the prompt includes examples with the same number of reasoning steps as those needed by the model.
>
> In our own model, Selection-Inference, the model only ever sees a single reasoning step at a time (but these can come from D0-D5 problems). An example prompt for depth-2 reasoning is shown in Section C.2. This works because (a) the inference step should be step-independent and (b) the Selection model is able to learn a heuristic about what rules and facts to choose based on the question and the statements available.
>
> The following is from Section C.2:
>
> “Below is an example selection prompt. Note that this is for a depth-2 problem and so we show examples of the first reasoning step where the conclusion would not directly prove or disprove the statement and the last reasoning step, where the conclusion would directly prove or disprove the statement.”

---

> > ### Comment · Reviewer_BSuH · 2022-11-21
> > **response to authors**
> >
> > Thank you for all the additional information you provided. This makes the paper complete and is an accept for me.

---

### Official Review · Reviewer_g5nq · 2022-10-21

**Confidence:** 5
**Clarity, Quality, Novelty And Reproducibility:** 1. It seems like the proposed method …
**Correctness:** 4
**Technical Novelty And Significance:** 4
**Empirical Novelty And Significance:** 3
**Recommendation:** 8

**Strength And Weaknesses:**


Strengths:
1.  The paper presents a comprehensive study of 46 reasoning tasks and gives an understanding of how good are the current LLMs in single step and multi-step reasoning problems.
2. The proposed selection-inference algorithm is a very interesting and novel idea. It makes the reasoning trace more interpretable.
3. The paper demonstrates that this approach will enable small language models to outperform baseline large language models that doesn’t use SI algorithm

Weaknesses:
1. It would have been a more interesting comparison to see how this method performs in comparison to other recent works such as self-consistency, verifiers, etc. The proposed method has its own merits, but comparing with more methods would have been more interesting.


**Summary Of The Paper:**

The paper explores the reasoning capabilities of large language models (LLMs). They carry out a comprehensive evaluation of 46 reasoning tasks to show that language models perform fairly well on single step reasoning problems, but suffer at multi-step reasoning problems. To that end, the paper proposes a new algorithm called selection-inference where LLMs solve multi-step reasoning problems by iterating over selection and inference modules to generate a series of interpretable causal reasoning steps. Through this method, they show that a smaller (in terms of parameters) LLM can outperform a baseline larger LLM. The paper also presents several interesting ablations with scale, fine-tuning, and performance on various levels of multi-step reasoning problems.

**Summary Of The Review:**

The paper provides some very interesting understanding of large language model capabilities for single vs. multi-step reasoning. The proposed SI algorithm is novel and very useful. All the ablations are very useful. Overall, I recommend this paper.

---

> ### Author Response · Authors · 2022-11-18
> **Thank you for reviewing our paper.**
>
> Thank you for reviewing our paper. We have addressed your questions below.
>
> > 1. It seems like the proposed method is computationally more heavy than CoT. Please have a discussion on that in the paper.
>
> We have added the following to Appendix K in the revised paper:
>
> "CoT involves a single forwards pass sampling `max_cot_tokens`. For a problem with H reasoning steps this would equate to roughly `cot_tokens_per_step = max_cot_tokens / H`.
>
> Selection-Inference on the other hand does not involve a single forward pass, but rather (scores and) samples sequentially. For simplicity, let us initially assume that in the selection step we sample tokens rather than score them. This means that we sample `H * (max_selection_tokens + max_inference_tokens)`. So in the case where  `cot_tokens_per_step = max_selection_tokens + max_inference_tokens` the number of computations is similar. This is a reasonable assumption since models were prompted to solve these problems in a similar way and we do not expect one model to solve problems with significantly more or less tokens than another model.
>
> Now consider that in the selection step, we do not sample tokens auto-regressively (as in CoT) but rather we score tokens. Tokens can be scored in parallel and therefore can be computed more efficiently than they can be sampled.
> Therefore, if implemented correctly, Selection-Inference can be at least as computationally efficient as CoT, if not more efficient."
>
> > 2. Does the paper see the chain-of-thought approach as not interpretable?
>
> The primary motivation for CoT is to improve final answer accuracy, rather than provide good reasoning traces. The authors of CoT show that their model is able to obtain the correct answer, via incorrect reasoning, suggesting that the reasoning trace does not always support the model’s final answer. So, although CoT traces may be interpretable, they are not as trustworthy as those of SI.
>
> > 3. Can you clarify on whether selection and inference use fixed prompts? If they are fixed, do you see any benefit in making them dynamic based on which iteration step of the reasoning the model is at?
>
> Thanks for this question. For selection, the prompt includes the latest context (with all previous inferences) and the question. This is sufficient information for the selection model. Providing more information, for example, the previous steps, would make the selection model “step” dependent which may affect generalisation to greater reasoning depths. By not appending the previous steps, the selection is independent of the reasoning step it is on, which is intended to help with generalisation.

---

> > ### Comment · Reviewer_g5nq · 2022-11-21
> > **Thanks for the response**
> >
> > I have read  author's response and also other reviewers' comments. I will keep my score same and recommend to accept this paper.

---

### Official Review · Reviewer_wTCG · 2022-10-22

**Confidence:** 4
**Correctness:** 4
**Technical Novelty And Significance:** 3
**Empirical Novelty And Significance:** 3
**Recommendation:** 8

**Clarity, Quality, Novelty And Reproducibility:**

### Questions and concerns:
1. The authors mention several times that the SI framework `is unlikely to make up information to answer the question`, I fail to understand why is that. It would be helpful to show some examples where COT or other systems make up information but SI does not, and discuss the reason.
2. It's not fully clear to me why in the selection step you have `Therefore, ` in your prompt. It seems to be the trigger of inference?
3. The authors very briefly mention in the conclusion section about `the halting issue`. I agree that this is one of the major limitations of this work. As the authors mention, this is the case for both 1) the fixed $H$ in Algorithm 1 and 2) the fixed $K$ in Algorithm 2. I wonder in the current version, are there any heuristics to prevent the LLM from receiving distracting information? This would happen when either $H$ or $K$ is greater than sufficient amount of evidence. Please elaborate a bit more on this, including how $H$ and $K$ are determined in the current version, and how the authors plan to make these numbers adaptable.
4. In Figure 4 (b), is there any specific reason that the 280B vanilla generative model performs so poorly on Babi tasks?


### Minor issues and typos:

1. Appendix A appears to be empty. I know it includes a bunch of figures, but it's a bit weird to be named `APPENDIX` and without any text content there.
2. I appreciate the authors include many examples and detailed prompt design in the appendix, however, the current version is not easy to read. A table of content would greatly increase the accessibility.
3. There are a few `see Appendix for ...` in the main body without specific appendix label, please add pointers.
4. Also accessibility issue, please increase font size for all text in Figure 3 and Figure 4. They are almost unreadable in a printed version.
5. Figure 1 caption: `Chain-of-Though` --> `Chain-of-Thought`.
6. The paragraph above Section 7: `... our approach is able to generalised beyond ...` --> `... our approach is able to generalise beyond ...`.


**Strength And Weaknesses:**

### Strength

* **Neat idea**: The idea of the two step selection-inference framework can be seen as a way to leveraging human prior knowledge in prompt engineering. As the authors discussed in the motivation section, this to some extent bridges neurosymbolic AI with recent large-scale deep learning approaches. There are a few neat ideas I like in this paper:

  1. Ranking negative log-likelihood for selection instead of directly prompting LLM for a list of outputs. This could effectively avoid the potential trouble of LLM generating something uncontrollable or hard to parse. Although I have to say this requires the LLM output probabilities to be available, which is not always the case (for practitioners who has less ML knowledge, or using online LLM APIs).
  2. "Occlude" question from the inference module. This is in some sense adding an information bottleneck.

* **Empirical value**: I'd love to see research along this line, which can potentially have great empirical value. Better prompting strategies can be useful in general and this is not limited to the ML community.
* **Significance test**: I appreciate the authors include significance test in their experiment and result section.
* **Interpretability, transparency, and humans-in-the-loop**: Compared to vanilla in-context learning and prior methods such as COT, the proposed framework can have a better interpretability and help researchers to better understand how an LLM makes certain decision. I also like the opportunity of humans-in-the-loop that the SI framework brings (e.g., users help to verify/filter selected evidence).
* **Explicit knowledge representation / memory**: In some sense, the iteratively growing set of evidence can be seen as a working memory, or a representation of past experience that one can easily retrieve from. Thinking broadly, from a sequential decision making or robotics perspective, instead of multi-step reasoning, I can imagine something like the SI framework being used to facilitate multi-step planning.


### Weaknesses

1. Please see my question 3 below regarding the $H$ and $K$ values.
2. The paper suffers from a few accessibility issues. Please see the minor issue and typos section below.

**Summary Of The Paper:**

In this work, the authors focus on leveraging pre-trained LLMs to solve language-based reasoning tasks, via in-context learning.

First, on a wide range of tasks, the authors investigate how well existing LLMs at various model sizes tackle logical reasoning tasks in a standard 5-shot prompting setting. Results suggest that they perform better on simple single step logical inference, but suffer from more complex tasks such as multi-step reasoning.

Motivated by the above observation, the authors propose a novel Selection-Inference (SI) framework, which is an iterative prompting method designed for solving logical reasoning tasks. At every iteration step, SI first ask the LLM to select a subset of most relevant supporting evidence from a larger set of input facts, conditioned on the question; then in an inference step, LLM is asked to infer a new fact from the selected evidence. The inferred new fact is either as the final answer, or added into the input fact set to the next iteration.

The authors show that equipped with this SI method, a LLM with 7B parameters can outperform a set of baselines, including a much larger 280B LLM, equipped with Chain of Thought (COT), the current state-of-the-art prompting method.

The authors also show that with some minor fine-tuning, a 7B LLM with SI can further a significant improvement.

The author also discuss and demonstrate how the SI framework can facilitate model interpretability, transparency, and humans-in-the-loop study.

**Summary Of The Review:**

Overall, I like this paper, it's clearly written, the proposed method makes sense to me and results suggest that the SI framework indeed improves the LLM's performance on a range of reasoning tasks. Unless I had some major misunderstandings, I recommend to accept this paper.

---

> ### Author Response · Authors · 2022-11-18
> **Thank you for your positive and detailed review (continued).**
>
>
> > 2. It's not fully clear to me why in the selection step you have Therefore, in your prompt. It seems to be the trigger of inference?
>
> The “Therefore” is only included in the ProofWriter prompts because this can help the model to learn how many statements should be selected. The reasons for this are discussed in Appendix D4 (see excerpt below):
>
> “Another challenge with the ProofWriter dataset is deciding how many arguments should be selected for each rule. In the ProofWriter dataset, some rules take one argument, others take two. We experimented with various different ways to encourage the model to stop selecting arguments. For example, we append ". Therefore, " as a choice to the context that the model can select. If the language model selects ". Therefore, " then the selection step ends. We allowed a maximum of two facts to be selected.”
>
> Note that when we use fine-tuned models and do selection by label, we no longer have this problem, but the “Therefore” is still a helpful indicator for when the selection step is complete.
>
> > 3. The authors very briefly mention in the conclusion section about the halting issue. I agree that this is one of the major limitations of this work. As the authors mention, this is the case for both 1) the fixed H in Algorithm 1 and 2) the fixed K in Algorithm 2. I wonder in the current version, are there any heuristics to prevent the LLM from receiving distracting information? This would happen when either H or K is greater than sufficient amount of evidence. Please elaborate a bit more on this, including how H and K are determined in the current version, and how the authors plan to make these numbers adaptable.
>
> What if K was bigger? We explored heuristics for when to stop selecting statements in Section D.4 (see the excerpt above).
>
> What if H was bigger? Figure A1 in Appendix D4 suggests that “too many” steps of reasoning do not lead to any degradation in performance. We have added the following discussion to D.4:
>
> "If the model is run for additional steps, once the answer has been reached, then the model often repeats its final reasoning step for the remainder of the iterations."
>
> In most cases, H and K are determined by the prompt. If we show examples of how to decompose the problem into H’ steps then we will use H=H’. Similarly, if the prompt includes K’ statements being selected on each iteration, we use K’=K. For most datasets the values of H and K do not vary within the dataset. ProofWriter if the exception and this is discussed in detail in Section D.4.
>
> Also note that in the fine-tuning setting, there is no K. We have updated Appendix D.4 in the revised paper with the following:
>
> “Note that when using fine-tuned models, estimating the number of statements via heuristics is no longer necessary since the model learns to generate a string with the required number of sentences referenced by sentence label. For example, `sent 1. We know that sent 7 and sent 5.`”
>
> > 4. In Figure 4 (b), is there any specific reason that the 280B vanilla generative model performs so poorly on Babi tasks?
>
> The reason for this apparent anomaly is that the 280B model tends to offer longer responses than the 7B model, with spurious extra text. This means that even when it outputs the correct answer, it is penalized for this additional text. In the light of your question, we re-ran the 280B model with some additional post-processing to remove the extra text. This does improve its average performance by 2.65%, and we have updated Figs.4a and 4b to show these more charitable numbers. The 280B model still performs very poorly compared to our approach, so none of our claims are significantly affected.
>
> > Minor issues and typos:
>
> Thank you for pointing these out. We have made the following changes and corrections in the revised paper.
> We have added text to Appendix A.
> Thanks for the great suggestion, we have included a contents page.
> We have provided specific references to the Appendix.
> We have fixed the typo in Section 7.
> We have increased the size of the figures. If the paper is accepted, we can further increase the figures for the camera ready version since we will have an additional page available.
> We have fixed the caption on Figure 1.

---

> > ### Comment · Reviewer_wTCG · 2022-11-18
> > **Thank you**
> >
> > I have read other reviewers' comments as well as the authors' response. They helped me to gain a better understanding of this work. I keep my score and I recommend to accept this paper.

---

> ### Author Response · Authors · 2022-11-18
> **Thank you for your positive and detailed review.**
>
> Thank you for your positive and detailed review. Please see our response below.
>
> > 1. The authors mention several times that the SI framework is unlikely to make up information to answer the question, I fail to understand why is that. It would be helpful to show some examples where COT or other systems make up information but SI does not, and discuss the reason.
>
> Consider this CoT example from Appendix J:\
> INPUT:\
> greg is a frog\
> greg is white\
> bernhard is a swan\
> lily is a lion\
> brian is a rhino\
> bernhard is yellow\
> brian is green\
> lily is yellow\
> julius is a frog\
> Question: what color is julius?\
> OUTPUT:  julius is a frog. We know that brian is a frog and that brian is green. Therefore, julius is green
>
> We have updated Appendix J in the revised paper to include following discussion:
>
> “The above example shows CoT making up a fact to answer the question. The question above asks “what color is julius?”. The CoT method correctly identifies that Julius is a frog. Then, rather than correctly saying “We know that greg is a frog” the model makes up a fact “brian is a frog” and combines this with the (existing) fact that “brian is green” to incorrectly conclude that “julius is green”. Our model, Selection-Inference, could not make this mistake, since the selection step is restricted to only selecting statements from the context (see Section 4.1).”

---

### Official Review · Reviewer_TQ9M · 2022-10-31

**Confidence:** 3
**Clarity, Quality, Novelty And Reproducibility:** The clarity is good, but novelty is s…
**Correctness:** 3
**Technical Novelty And Significance:** 3
**Empirical Novelty And Significance:** 3
**Recommendation:** 6

**Strength And Weaknesses:**

trengths

- This framework of SI is different from all previous frameworks, as illustrated in Figure 1. SI use LLMs to produce each reasoning step one at a time, which is similar with ProofWriter. While ProofWriter can only answer “Prove this statement to be True/False” style question, SI is able to solve reasoning problems.

- The designs of splitting of each step of reasoning into selection and interference makes the model more unlikely to make up information to answer and it cannot ignore previous reasoning when getting the answer.

- The authors conduct extensive experiment and provide a comprehensive evaluation of LLMs on a set of 46 tasks. They show that LLMs are good at simpler single step logical inference in 5-shot generalisation settings, but struggle with harder problems.

- Using SI framework, the 7B model shows impressive results and even outperform 280B LLM baseline using COT framework. Also, the SI framework has ability to produce a causal reasoning trace and is able to recover from errors.

Weaknesses

- Some important details are not presented. The algorithm now just simply halts after some times of steps. So will the result differs from different numbers of steps, and how.

- The results are not convincing enough. In the paper, the authors just show the results of 7B SI. I wonder know that if the 280B version using SI will outperform all previous models.

- There are many case studies, more quantitative metrics to evaluate the effect is better.

- There are some typos in this paper, which causes additional burden for understanding. For example:

1) typo in page 2, “Figure 1” paragraph, “indicate” -> “indicates”

2) typo in page 5, “We use prompt...” paragraph, “the the following form” -> “the following form”

3) typo in page 26, “We have seen...” paragraph, “which we we now” -> “which we now”

4) typo in page 26, “When observing...” paragraph, “we us” -> “we use”

**Summary Of The Paper:**

The paper proposes a novel framework called Selection-Inference(SI), which exploits pre-trained large language models as general processing modules to solve logical reasoning problems. Using this framework, LLMs will continuously alternates between selection step and inference step to generate a sequence of casual reasoning steps. So not only the answers can be concluded from these reasoning steps, but also these causal natural-language-based reasoning trace helps us to learn how the model reached the answer and opens the system’s decisions to human critique. The results are very impressive. The paper shows that a 7B SI model outperforms the 280B model.

**Summary Of The Review:**

This paper proposes SI framework, which improves the ability of LLMs to solve reasoning problems and outperforms other framework including COT. Even though this framework still has many limitations, it does develop a new approach and achieves state-of-the-art results. And this work can be extended to many further works, which are inspirational for other works in the field.

---

> ### Author Response · Authors · 2022-11-18
> **Thank you for reviewing our paper. We have addressed your concerns below:**
>
> Thank you for reviewing our paper. We have addressed your concerns below:
>
> > Some important details are not presented. The algorithm now just simply halts after some times of steps. So will the result differs from different numbers of steps, and how.
>
> In Section D.4, Figure A1, in the appendix, we show an example of how the number of reasoning steps affects the final answer accuracy.
>
> We have updated Section D.4 to include the following observation:
> “If the model is run for additional steps, once the answer has been reached, then the model often repeats its final reasoning step for the remainder of the iterations.”
>
> > There are many case studies, more quantitative metrics to evaluate the effect is better.
>
> Our main quantitative results are in Figures 3, 4 and 6. Additional quantitative results can be found in Figures A5, A6, and Table A1.
> * In Figure 3 we evaluate three language models of different sizes on 56 tasks.
> * A per-task breakdown is shown in Figure A5.
> * In Figure 4 we compare final answer accuracy of Selection inference both with prompt engineering and fine-tuning to 6 baseline models over 7 tasks.
> * In Figure 6 we evaluate the quality of the reasoning trace on ProofWriter. This was only possible for ProofWriter because the other datasets do not provide ground truth reasoning traces.
> * Figure A6 compares the proportion of problems on which SI and COT make up facts.
> * Table A1 shows a human evaluation comparing SI to COT.
>
> > Typos
>
> Thanks for pointing these out, we have fixed these in the revised paper.

---

### Official Review · Reviewer_QvM7 · 2022-10-31

**Confidence:** 4
**Clarity, Quality, Novelty And Reproducibility:** Detailed in the main review.
**Correctness:** 3
**Technical Novelty And Significance:** 2
**Empirical Novelty And Significance:** 3
**Recommendation:** 8

**Strength And Weaknesses:**

## Strengths
- ****************Clarity:**************** The paper is well-written and easy to follow. The problem, and motivation are clear and the method directly follows from them.
- **************Novelty:************** Splitting reasoning into modules has been explored but to my knowledge, using something like the proposed SI method has not been seen.
- The SI framework significantly improves performance on the reasoning tasks over vanilla CoT/ scratchpad approaches.
- Selecting a subset of information ensures that the language model is restricted by the information it uses for reasoning.
## Weaknesses
- One missing baseline for the framework is where the CoT prompts are framed as SI prompts without splitting SI into two separate modules.
- SI using larger language models: I expected to see how models a magnitude larger than 7B would perform with the SI framework. This would help us understand how important the specific components of the framework are.
- For example, a larger model might not make facts up in the selection step when using greedy decoding (with temperature = 0).
- Or, one might not need two separate calls to the LM for S and I, simply using a CoT script with steps for selection and inference is all that one might need.

### Other

- **********************************Reproducibility:********************************** The method is straightforward to implement and test. The authors provide the prompts but access to the larger 280B language model is restricted and the baseline results would be impossible to replicate.

**Summary Of The Paper:**

The paper introduces a new framework for multi-step reasoning by splitting reasoning into two main stages: selection and inference. Selection involves choosing relevant information from the question to the inference module. The inference module uses the information from the selection module to reason and produce an answer. The authors test the framework on 10 logical reasoning tasks and show that they can get a 7B parameter lm to outperform 280B parameter lm.

**Summary Of The Review:**

The approach is empirically quite strong! The paper would be better with some additional baselines and justifications for design choices.

---

> ### Author Response · Authors · 2022-11-18
> **Thank you for providing a positive review for our paper.**
>
> Thank you for providing a positive review for our paper.
>
> > One missing baseline for the framework is where the CoT prompts are framed as SI prompts without splitting SI into two separate modules.
>
> The CoT prompts are already framed as SI prompts but without splitting into two modules. Examples of this are shown in Appendix C. To make this clear, we have included the following in the revised paper:
>
> “We also consider a chain-of-thought (COT) (Wei et al. 2022) inspired baseline, where the k-shot prompts to the 7B and 280B models include reasoning traces for the same examples that we use to prompt the SI framework (although with selection and inference combined, see Appendix C for example prompts).”

---

### Decision · Program_Chairs · 2023-01-20

**Decision:**

Accept: notable-top-5%

**Justification For Why Not Higher Score:**

N/A

**Justification For Why Not Lower Score:**

All reviewers agree with the novelty, clarity, and solid experimental results of this paper.

**Metareview: Summary, Strengths And Weaknesses:**

The paper proposes a new framework for multi-step logical reasoning. It includes two stages: selection and inference. In particular they first select a few pieces of information by scoring each fact in the context with the model’s log-likelihood. Then the retrieved facts are combined and the model generates a conclusion based only on those facts. The authors test the framework on 10 logical reasoning tasks and show that their 7B parameter LM outperforms 280B parameter LM.

Strength of the paper:
1. Novel framework of Selection/Inference for multi-step reasoning.
2. Extensive experiments and strong results.
3. The 7B model outperforms 280B baseline.

Weakness of the paper:
1. some technical details  are unclear
2. more explanation of prompts are needed.
3. Selection of H and K

Overall, this paper has adequate novel technical contribution and strong results on a variety of reasoning tasks.

**Note From Pc:**

if the above contains the word "oral" or "spotlight" please see: "oral" presentation means -> notable-top-5% and "spotlight" means -> notable-top-25%. As stated in our emails, we are disassociating presentation type from AC recommendations

---

> ### Author Response · Authors · 2023-03-01
> **Camera ready version of the paper addresses the weaknesses above.**
>
> To address the weaknesses noted above, we have included our prompts in the supplementary material and have included Figure A1 which shows the effects of additional reasoning steps, H. Additional discussion of H and K has been added to Section D.4.